# TimeInf: Time Series Data Contribution via Influence Functions

**Yizi Zhang**[1][†][*]  **Jingyan Shen**[1][*]  **Xiaoxue Xiong**[1][*]  **Yongchan Kwon**[1][†]

[1]Columbia University

## Abstract

Evaluating the contribution of individual data points to a model's prediction is critical for interpreting model predictions and improving model performance. Existing data contribution methods have been applied to various data types, including tabular data, images, and text; however, their primary focus has been on i.i.d. settings. Despite the pressing need for principled approaches tailored to time series datasets, the problem of estimating data contribution in such settings remains under-explored, possibly due to challenges associated with handling inherent temporal dependencies. This paper introduces **TimeInf**, a model-agnostic data contribution estimation method for time-series datasets. By leveraging influence scores, TimeInf attributes model predictions to individual time points while preserving temporal structures between the time points. Our empirical results show that TimeInf effectively detects time series anomalies and outperforms existing data attribution techniques as well as state-of-the-art anomaly detection methods. Moreover, TimeInf offers interpretable attributions of data values, allowing us to distinguish diverse anomalous patterns through visualizations. We also showcase a potential application of TimeInf in identifying mislabeled anomalies in the ground truth annotations.

## 1 Introduction

Understanding how individual data points impact model outcomes is a crucial task, especially in fields like finance and healthcare, where the decision-making process depends heavily on model predictions. By evaluating the influence of training data points, we can identify outliers or low-quality samples that deviate from the normal behavior or significantly degrade the model performance. Moreover, the identification of influential data points allows researchers and practitioners to focus on meaningful patterns in their datasets. From all these motivations, the attribution of a value to each data point—distinguishing between beneficial and detrimental patterns—is of critical importance to various applications, including anomaly detection and the recognition of valuable data patterns (Kwon & Zou, 2023; Sun et al., 2025).

A standard data attribution approach is the regular influence function, originally developed in robust statistics (Hampel, 1974; Cook & Weisberg, 1980). It measures the change in model parameters or predictions in response to infinitesimal adjustment in the weight of a specific data point, effectively capturing the impact of individual data points. However, its underlying assumption of independent and identically distributed (i.i.d.) data limits its direct application to time series datasets, due to the temporal dependency within data points. To address this, Kunsch (1984) introduces a conditional influence measure for autoregressive (AR) models, which quantifies the impact of a single time point given the values of its immediate predecessors. Although this approach extends and validates the use of the regular influence function in time series through some preprocessing, its dependency on past data limits its capability to express a data point's influence within specific historical contexts and fails to fully capture the influence. Alternatively, Ghosh et al. (2020) employs approximate leave-one-out cross-validation to quantify the effect of removing a single observed time point on model performance while preserving temporal dependencies in the latent space. This method gives a rigorous attribution method for time series data, however, it is strictly limited to specific latent state-space models.

---

[*]Equal contribution    [†] Corresponding author (e-mail: yz4123@columbia.edu; yk3012@columbia.edu)

We propose a new approach called TimeInf that addresses the challenges of existing time series data attribution methods. Our method integrates the influences over multiple sampled time blocks (Hall, 1985; Kunsch, 1989; Bühlmann, 2002) that include the particular time point of interest. Unlike the conditional influence of Kunsch (1984), which evaluates a time point's impact based on its immediate past, our distinctive integration considers various temporal patterns and accurately reflects the true effect of each data point. Moreover, unlike Ghosh et al. (2020), our method does not require specific model structures.

We demonstrate its effectiveness in time series anomaly detection. On multiple real-world datasets, our method outperforms existing approaches in detecting harmful anomalies. TimeInf provides intuitive data value attributions, facilitating the identification of harmful anomalies and beneficial temporal patterns through visualizations of the computed influence scores. We present a potentially important application of TimeInf in identifying mislabeled anomalies in the ground truth annotations. In Appendix, we demonstrate that TimeInf is model-agnostic and can distinguish between beneficial and detrimental training data for time series forecasting.

## 2 PRELIMINARIES

**Notations.** We denote a set of time series observations by $\{x_i\}_{i=1}^n$ where $x_i \in \mathbb{R}$ are drawn from a stationary ergodic process (Edition et al., 2002). We construct $n - m + 1$ $(m < n)$ overlapping time blocks of consecutive observations $x_i^{[m]} := (x_i, \ldots, x_{i-m+1}) \in \mathbb{R}^m$ with the empirical $m$-dimensional marginal distribution[1] $\hat{F}^m$ defined as

$$\hat{F}^m = \frac{1}{n - m + 1} \sum_{i=m}^n \delta_{x_i^{[m]}},$$

where $\delta_{x^{[m]}}$ is the point mass distribution at $x^{[m]}$, and $\hat{F}^m \in \mathcal{M}_{\text{stat}}^m$, where $\mathcal{M}_{\text{stat}}^m$ is the $m$-dimensional marginals of stationary process.

**Influence Functions.** One way to introduce influence functions for autoregressive models is through the concept of a contaminated distribution function. For $F^m \in \mathcal{M}_{\text{stat}}^m$, $0 < \epsilon < 1$, and $z^{[m]} \in \mathbb{R}^m$, the contaminated distribution function is defined as follows.

$$F_\epsilon^m(z^{[m]}) = (1 - \epsilon)F^m + \epsilon\delta_{z^{[m]}}.$$

Intuitively, $F_\epsilon^m(z^{[m]})$ is a contaminated distribution in which the original distribution $F^m$ is contaminated by a sample $z^{[m]}$ with a probability of $\epsilon$.

Denote $\theta \in \Theta \subseteq \mathbb{R}^q$ as the time series model parameters, and $\hat{\theta}$ as an estimator. We introduce a general definition of the influence function for time series models as follows.

**Definition 2.1** (**Influence Function for $m$-Dimensional Marginal**). *The influence function of a sample $z^{[m]}$ (or equivalently $\delta_{z^{[m]}}$) on an estimator $\hat{\theta} : \mathcal{M}_{\text{stat}}^m \to \mathbb{R}^q$ at a distribution $F^m$ is defined as follows.*

$$\mathcal{I}_{\hat{\theta}}(\delta_{z^{[m]}}, F^m) = \lim_{\epsilon \searrow 0} \frac{\hat{\theta}(F_\epsilon^m(z^{[m]})) - \hat{\theta}(F^m)}{\epsilon}. \tag{1}$$

That is, influence functions measure the infinitesimal effect of upweighting $z^{[m]}$ by $\epsilon$ to $\hat{\theta}$ evaluated at $F^m$. We set $\mathcal{I}_{\hat{\theta}}(\delta_{z^{[m]}}, F^m) = \mathcal{I}_{\hat{\theta}}(z^{[m]})$ when the second argument $F^m$ is an empirical distribution $\hat{F}^m$, and use a simpler form $\mathcal{I}_{\hat{\theta}}$ when the context is clear.

**Example 1** (*$M$-estimators*). Equation (1) depends on a specific choice of $\hat{\theta}$. Given that most machine learning estimators are based on optimizing a criterion function, we focus on a broad class of these estimators, which is also known as $M$-estimators (Hampel, 2001). Suppose a loss function

---

[1]We focus on univariate data for notation simplicity, but the concepts can be readily extended to multivariate settings.

Table 1: *Comparative overview of time series data contribution estimation methods. Data Instance*: the specific instance whose influence on the model we aim to quantify. *Quantity of Interest*: the model output or characteristic that changes in response to changes in the data instance. *Contaminating Distribution*: the distribution of $z^{[m]}$, introduced to perturb the original distribution $F^m$. *Model Choice*: the range of model types compatible with the method.

| Method | Data Instance | Quantity of Interest | Contaminating Distribution | Model Choice |
|---|---|---|---|---|
| Regular Influence (Koh & Liang, 2017) | I.I.D. | Loss | $\delta_{z^{[m]}}$ | Differentiable Model |
| Conditional Influence (Kunsch, 1984) | Time Point | Parameter | $\delta_{z^{[m]}}$ | Autoregressive Model |
| LWCV (Ghosh et al., 2020) | Time Point | Loss | $\delta_{z^{[m]}}$ | Differentiable Latent State-Space Model |
| TimeInf (Ours) | Time Point | Loss | Mixture of $\delta_{z^{[m]}}$ | Model-Agnostic |

$\rho : \mathbb{R}^m \times \Theta \to \mathbb{R}$ is differentiable with respect to $\theta$ and strongly convex. We define $M$-estimators as the solution to the following minimization problem.[2]

$$\hat{\theta} = \underset{\theta \in \Theta}{\operatorname{argmin}} \int \rho(x^{[m]}, \theta) d\hat{F}^m(x^{[m]}), \tag{2}$$

which can be expressed as the solution to a set of equations

$$\int \psi(x^{[m]}, \theta) d\hat{F}^m(x^{[m]}) = 0, \quad \psi(x^{[m]}, \theta) := \frac{d\rho(x^{[m]}, \theta)}{d\theta}.$$

This class includes the maximum likelihood estimator in AR models (Davis et al., 1992; Bai & Wu, 1997). In practice, with mild regularity conditions (Godambe, 1960), the empirical influence function (Cook & Weisberg, 1980; Koh & Liang, 2017) of $z^{[m]}$ on $\hat{\theta}$ at $\hat{F}^m$ is given as follows.

$$\mathcal{I}_{\hat{\theta}}(z^{[m]}) = - \left( \int \frac{d\psi(x^{[m]}, \hat{\theta})}{d\theta} d\hat{F}^m(x^{[m]}) \right)^{-1} \psi(z^{[m]}, \hat{\theta}). \tag{3}$$

See Appendix Section A for derivations of Equation (3) for detailed proof.

## 2.1 PRIOR WORKS IN DATA ATTRIBUTION

Equation (1) presents a general definition of influence functions, encompassing many existing influence function-based methods as special cases. This section briefly summarizes the strengths and limitations of existing works, paving the way for a new framework to assess data contribution in time series settings.

**Regular Influence.** The regular influence function is originally proposed by Hampel (1974) to measure a data instance's impact on model parameters and later repurposed by Koh & Liang (2017) to assess the impact on test predictions. It is a special case of the general influence function formulation in Equation (1) when $m = 1$. It can be applied to any differentiable models, providing rigorous and intuitive explanations for identifying which data points are helpful or harmful. However, its limitation to $m = 1$ does not extend to general time series cases. For instance, the objective function for an AR model of order 1 requires two consecutive data points, $x_i^{[2]} = (x_i, x_{i-1})$, in a loss function, *i.e.*, $\rho(x_i^{[2]}, \theta) = (x_i - x_{i-1}\theta)^2$. This is not feasible when $m = 1$ because a loss function is evaluated at individual data points. We generalize this framework by considering any $m$ and construct rigorous influence functions for time series data.

**Conditional Influence.** There has been an approach to apply the regular influence function to a block of contiguous time series data points. While this seems a naive extension of the regular influence function, it has not been clear until Kunsch (1984) establishes its theoretical support for AR models. Specifically, Kunsch (1984) considers a conditional log likelihood (e.g., $\log p(x_{i+1} \mid x_i, \ldots, x_{i-m+1})$) for a loss function $\rho$ in Equation (2) and defines the conditional influence of $x_{i+1}$ given its immediate predecessors $\{x_i, \ldots, x_{i-m+1}\}$ by leveraging the regular influence function. Note that this method is also a special case of Equation (1) when $m > 1$.

---

[2] It is noteworthy that Equation (2) is a general version of the standard $M$-estimator (Hampel, 2001) in a sense that each loss function term $\rho(x_i^{[m]}, \theta)$ is evaluated at multiple data points $x_i^{[m]} = (x_i, \ldots, x_{i-m+1})$. When $m = 1$, *i.e.*, an individual loss is evaluated at a data point, Equation (2) yields the regular $M$-estimator.

Although Kunsch (1984)'s method measures a time point's impact on model parameters, it constrains the influence to a specific temporal arrangement as it assigns a point mass to its observed predecessors. In other words, it only considers the local historical context of the time point of interest, and as a result, it cannot account for the diverse contamination patterns that $\{x_i, \ldots, x_{i-m+1}\}$ can exhibit. We address this problem by considering a more informative contaminating distribution.

**LWCV.** The leave-within-structure-out cross-validation (LWCV) method, introduced by Ghosh et al. (2020), measures the impact of removing a single time point on model loss without breaking the temporal dependency within the data. In contrast to existing influence function variants, which measure the impact on model parameters, LWCV captures the impact on model loss, providing additional insights into how data are influencing models. However, this approach is specifically designed for state-space models such as autoregressive hidden Markov models (AR-HMM) to maintain temporal dependencies through latent variables. This underlying assumption on latent structures restricts LWCV's applicability to models with explicit latent states, excluding more complex models like transformers.

## 3    TimeInf: Time Series Data Contribution via Influence Functions

We provide a unified framework for time series data contribution that addresses the limitations of the aforementioned approaches. Throughout this section, we focus on differentiable models to ease the presentation of our methods, but the proposed method can be applied to non-differentiable models. For the treatment of non-differentiable models, see Appendix Section B.

The general influence function $\mathcal{I}_{\hat{\theta}}$ defined in Equation (1) captures how data contamination on a time block impacts model parameters. To understand a specific time block's influence on model predictions instead, we can use the chain rule to derive the influence function for a time block $z^{[m]}$:[3]

$$
\begin{aligned}
\mathcal{I}_{\text{block}}(z^{[m]}, z_{\text{test}}^{[m]}) &= -\psi(z_{\text{test}}^{[m]}, \hat{\theta})^\top \mathcal{I}_{\hat{\theta}}(z^{[m]}) \\
&= -\psi(z_{\text{test}}^{[m]}, \hat{\theta})^\top \left( \int \frac{d\psi(x^{[m]}, \hat{\theta})}{d\theta} d\hat{F}^m(x^{[m]}) \right)^{-1} \psi(z^{[m]}, \hat{\theta}),
\end{aligned}
\tag{4}
$$

which measures the impact of overweighting $z^{[m]}$ on model prediction based on a test sample $z_{\text{test}}^{[m]}$. This can be viewed as either a straightforward extension of the regular influence function (Koh & Liang, 2017) or that of the conditional influence function (Kunsch, 1984). A critical limitation of this approach is its inability to quantify influence scores for individual data points, as it is defined for a block of contiguous data points. Moreover, the evaluation of the influence function constraints to a specific temporal arrangement $z^{[m]}$ because it only considers a point mass to the observed $z^{[m]}$.

To address these challenges, we propose to consider all contiguous time blocks $z^{[m]}$ that contain a specific data point $z$ and aggregate them into a single value. This idea is formalized into an integration of block influence functions over an empirical contaminating process:

**Definition 3.1 (TimeInf).** *Let $S_{z, \hat{F}^m}$ be the set of overlapping time blocks $z^{[m]} \sim \hat{F}^m \in \mathcal{M}_{stat}^m$ containing time point $z$. For $\hat{G}^m = \frac{1}{|S_{z, \hat{F}^m}|} \sum_{z^{[m]} \in S_{z, \hat{F}^m}} \delta_{z^{[m]}}$, we define TimeInf as the integration of $\mathcal{I}_{block}(z^{[m]}, z_{test}^{[m]})$ with respect to $z^{[m]} \sim \hat{G}^m$:*

$$
\mathcal{I}_{\text{time}}(z, z_{\text{test}}^{[m]}) := \int \mathcal{I}_{\text{block}}(z^{[m]}, z_{\text{test}}^{[m]}) d\hat{G}^m(z^{[m]})
\tag{5}
$$

$$
= -\psi(z_{\text{test}}^{[m]}, \hat{\theta})^\top \left( \int \frac{d\psi(x^{[m]}, \hat{\theta})}{d\theta} d\hat{F}^m(x^{[m]}) \right)^{-1} \int \psi(z^{[m]}, \hat{\theta}) d\hat{G}^m(z^{[m]}).
\tag{6}
$$

TimeInf $\mathcal{I}_{\text{time}}(z, z_{\text{test}}^{[m]})$ aggregates block influence functions $\mathcal{I}_{\text{block}}$ over various contaminating blocks defined in $S_{z, \hat{F}^m}$. We find this integration to be critically important as multiple temporal arrangements considered in $S_{z, \hat{F}^m}$ lead to a more robust influence measure that is less dependent on any single block

---

[3]Derivations of Equation (4) are included in Appendix Section A.

$\delta_{z^{[m]}}$, unlike Kunsch (1984)'s method. This advantage becomes clearer in an alternative expression of TimeInf. A simple algebra gives:

$$\mathcal{I}_{\text{time}}(z, z_{\text{test}}^{[m]}) = \lim_{\epsilon \searrow 0} \frac{\mathbb{E}\left[\rho(z_{\text{test}}^{[m]}, \hat{\theta}(\hat{F}_{\epsilon}^m(z^{[m]}))) - \rho(z_{\text{test}}^{[m]}, \hat{\theta}(\hat{F}^m))\right]}{\epsilon}, \tag{7}$$

where $\rho$ is the loss function and the expectation is taken over $\hat{G}^m$. This implies that TimeInf can be interpreted as the limit of expected infinitesimal change when a perturbation $\delta_{z^{[m]}}$ is introduced randomly with a contaminating distribution $\hat{G}^m$. Compared to existing influence function variations, the expectation in Equation (7) effectively captures the real impact of a per-time data point, avoiding the randomness of a particular block.

In summary, TimeInf is designed for time series data contribution and improves upon existing methods in the following ways:

- Unlike the regular influence function, TimeInf can capture per-time-point influence on specific model predictions while preserving the dependency structure between time points.
- In contrast to the conditional influence, TimeInf extends beyond the observed local historical context of the time point of interest.
- Unlike LWCV, TimeInf is model-agnostic, extending its applicability to complex models without explicit latent states.

### 3.1 VARIANTS OF TIMEINF

Depending on the task, different TimeInf variants can be used for data contribution estimation. When the goal is to estimate data quality, particularly in anomaly detection, self-influence provides a means to quantify the impact of a specific time point on its own prediction:

$$\mathcal{I}_{\text{self}}(z) = \frac{1}{|S_{z,\hat{F}^m}|} \sum_{z^{[m]} \in S_{z,\hat{F}^m}} \mathcal{I}_{\text{block}}(z^{[m]}, z^{[m]}), \tag{8}$$

The key idea is that anomalies, being rare and distinct, are hard to predict using normal data but easier using anomaly data itself. Thus, anomaly points are expected to be most influential for their own predictions.

For time series forecasting, to understand how changing a time point affects test predictions, we can average $\mathcal{I}_{\text{time}}$ from Equation (6) across all test set time blocks:

$$\mathcal{I}_{\text{test}}(z) = \frac{1}{|S_{\text{test},\hat{F}^m}|} \sum_{z_{\text{test}}^{[m]} \in S_{\text{test},\hat{F}^m}} \mathcal{I}_{\text{time}}(z, z_{\text{test}}^{[m]}), \tag{9}$$

where $S_{\text{test},\hat{F}^m}$ is the set of time blocks in the test set.

**Example 2** (TimeInf for AR($m$) model). Although any differentiable models can be used to compute TimeInf, our primary focus in this paper is on linear AR models due to their powerful performance and computational efficiency (Zeng et al., 2023; Toner & Darlow, 2024). For linear AR models of order $m$ and the mean squared error (MSE) loss $\rho(x_{i+1}^{[m+1]}, \theta) = (x_{i+1} - x_i^{[m]\top}\theta)^2$, the $M$-estimator $\hat{\theta}$ corresponds the least squares estimator. In this case, $\mathcal{I}_{\hat{\theta}}$ in Equation (3) and $\mathcal{I}_{\text{block}}$ in Equation (4) have closed-form solutions. Specifically, the influence of overweighting $z_l^{[m]}$ on the model parameters is given by:

$$\mathcal{I}_{\hat{\theta}}(z_l^{[m]}) = \left(\frac{1}{n}\sum_{i=1}^{n} x_i^{[m]} x_i^{[m]\top}\right)^{-1} z_l^{[m]}(z_{l+1} - z_l^{[m]\top}\hat{\theta}).$$

The influence of overweighting $z_l^{[m]}$ on the test prediction is:

$$\mathcal{I}_{\text{block}}(z_l^{[m]}, z_{\text{test},r}^{[m]}) = -(z_{\text{test},r+1} - z_{\text{test},r}^{[m]\top}\hat{\theta})^{\top} z_{\text{test},r}^{[m]\top} \left(\frac{1}{n}\sum_{i=1}^{n} x_i^{[m]} x_i^{[m]\top}\right)^{-1} z_l^{[m]}(z_{l+1} - z_l^{[m]\top}\hat{\theta}),$$

which can be plugged into Equation (5) to obtain TimeInf.

## 3.2 IMPLEMENTATION OF TIMEINF

The challenge in computing TimeInf lies in the inversion of the Hessian in Equation (4), expressed as $\int \frac{d\psi(x^{[m]}, \hat{\theta})}{d\theta} d\hat{F}^m(x^{[m]})$. With $n$ training points and $\theta \in \mathbb{R}^q$, this involves $O(nq^2 + q^3)$ operations. While it is feasible to directly compute the inverse of the Hessian for linear AR models with a small $q$, for deep neural networks with millions of parameters, this computation becomes prohibitively expensive. To approximate this computation efficiently, we resort to the conjugate gradient (CG) method, as suggested by Koh & Liang (2017), and the Hessian-free approach introduced by Pruthi et al. (2020). The CG method converts the matrix inversion into an optimization problem solvable in $O(nq)$ time. The Hessian-free approach replaces the inverse Hessian matrix with an identity matrix, reducing the main computation bottleneck of the influence function to only the dot product of gradients. Once equipped with the influence of a time block ($\mathcal{I}_{\text{block}}$), we can compute TimeInf ($\mathcal{I}_{\text{time}}$), self-influence ($\mathcal{I}_{\text{self}}$), and the influence of a time point on the test prediction ($\mathcal{I}_{\text{test}}$), according to Equations (6), (8), and (9).

## 4 USE CASES OF TIMEINF

This section investigates the practical effectiveness of TimeInf in real-world anomaly detection tasks. Our method outperforms existing time series data attribution methods as well as state-of-the-art anomaly detection methods. In addition, we illustrate how TimeInf facilitates the detection of various anomaly patterns through interpretable visualizations. Furthermore, we highlight the capability of TimeInf to identify mislabeled anomalies within ground truth annotations. We include additional experiments in Appendices C and D to demonstrate the model-agnostic nature of TimeInf, and how different hyperparameters (*e.g.*, block length and model size) affect the downstream task performance.

### 4.1 TIME SERIES ANOMALY DETECTION

When a time series model is trained with outliers or anomalous patterns, the quality of its predictions is expected to be negatively affected by these samples. In such situations, evaluating the impact of individual data points can be potentially useful in detecting anomalies. Motivated by this, we conduct time series anomaly detection experiments and assess the detection ability of TimeInf.

**Experimental Settings.** We consider realistic and practical experimental settings where the training dataset is not necessarily anomaly-free, and the proportion of anomalies is unknown (Jiang et al., 2022; Schmidl et al., 2022). We use the contaminated test set as training data for all methods and evaluate their performance on the same dataset. It is noteworthy that our setting differs from conventional time series anomaly detection settings, which often focus on detecting anomalies in a contaminated *test* dataset after training models on a well-curated clean training dataset (Alnegheimish et al., 2023; Si et al., 2024). While we recognize that each setting has its advantages, datasets with anomalous data points have been considered to investigate the detection ability of data attribution methods. That is, we want to know which training data points are anomalies and attempt to detect them using data contribution methods. We provide implementation details in Appendix Section F.

**Datasets.** We extensively evaluate TimeInf on five benchmark datasets: UCR (Wu & Keogh, 2021), SMAP (Hundman et al., 2018), MSL, NAB-Traffic, and SMD (Su et al., 2019). Detailed descriptions of these datasets and results on additional datasets are provided in Appendix Section F.

**TimeInf Usage.** We use self-influences $\mathcal{I}_{\text{self}}$ for anomaly detection. For univariate time series, we compute the self-influence directly for each time point, normalizing the results to a range between 0 and 1 to obtain anomaly scores. In the case of multivariate time series, we first apply self-influence to each dimension separately and then average the anomaly scores across all dimensions to obtain the final anomaly scores for individual time points. Although we can apply self-influences to all dimensions of the multivariate time series and obtain one set of anomaly scores, empirical results suggest that it is less effective than taking the average of anomaly scores obtained separately from each dimension. We provide detailed discussion in Appendix Section G.

**Baselines.** We compare TimeInf to three time series data contribution methods: block leave-one-out-cross-validation (LOOCV), conditional influence (Kunsch, 1984), and LWCV (Ghosh et al., 2020). Conditional influence and LWCV are described in Section 2.1. We have proposed Block LOOCV as a straightforward approach for calculating a time point's influence. This method removes all time

Table 2: *Anomaly detection performance and time cost of TimeInf and baseline methods. For both F1 and AUC metrics, a higher value indicates a better performance. TimeInf outperforms state-of-the-art time series anomaly detection methods (shown in white) and other data contribution techniques (shown in grey) across five real-world datasets, while being computationally efficient. Methods marked by "–" failed to complete experiments on certain datasets due to computational inefficiency.*

| | UCR | | | SMAP | | | MSL | | | NAB-Traffic | | | SMD | | |
|---|---|---|---|---|---|---|---|---|---|---|---|---|---|---|---|
| | AUC | F1 | Time (s) | AUC | F1 | Time (s) | AUC | F1 | Time (s) | AUC | F1 | Time (s) | AUC | F1 | Time (s) |
| Isolation Forest | 0.57 | 0.02 | 0.57 | 0.67 | 0.33 | 0.14 | 0.68 | 0.31 | 0.11 | 0.57 | 0.20 | 0.22 | 0.82 | 0.37 | 0.30 |
| LSTM | 0.60 | 0.04 | 803.27 | 0.55 | 0.31 | 245.52 | 0.70 | 0.34 | 8.77 | 0.57 | 0.19 | 49.13 | 0.79 | 0.30 | 307.08 |
| ARIMA / VAR | 0.54 | 0.10 | 2598.22 | 0.58 | 0.26 | 20.43 | 0.62 | 0.24 | 6.85 | 0.55 | 0.20 | 17.52 | 0.71 | 0.16 | 6.38 |
| TranAD | 0.50 | 0.22 | 2.87 | 0.55 | 0.21 | 0.84 | 0.40 | 0.06 | 0.24 | 0.60 | 0.36 | 1.44 | 0.65 | 0.05 | 3.54 |
| DCdetector | 0.74 | **0.34** | 151.08 | 0.67 | 0.35 | 194.52 | 0.69 | 0.34 | 50.13 | 0.61 | 0.32 | 27.96 | 0.70 | 0.25 | 106.70 |
| Anomaly Transformer | 0.50 | 0.01 | 471.96 | 0.49 | 0.04 | 51.24 | 0.42 | 0.10 | 20.13 | 0.46 | 0.05 | 74.86 | 0.51 | 0.05 | 246.80 |
| Block LOOCV | – | – | – | 0.71 | 0.32 | 215.83 | 0.70 | 0.29 | 30.93 | 0.62 | 0.33 | 18.69 | 0.85 | **0.42** | 15364.52 |
| Conditional Influence | 0.65 | 0.32 | 9.09 | 0.70 | **0.37** | 7.78 | 0.69 | 0.32 | 4.06 | 0.61 | 0.29 | 6.50 | 0.85 | 0.33 | 137.26 |
| LWCV | 0.55 | 0.03 | 1540.47 | 0.63 | 0.26 | 354.30 | 0.65 | 0.32 | 109.84 | 0.59 | 0.37 | 67.10 | 0.79 | 0.29 | 3369.52 |
| **TimeInf (Ours)** | **0.79** | 0.23 | 23.61 | **0.73** | 0.34 | 10.59 | **0.72** | **0.35** | 6.17 | **0.64** | **0.39** | 2.79 | **0.87** | 0.25 | 20.85 |

blocks containing the time point of interest, and then measures the impact of this removal on model losses; see Appendix Section F for implementation details about each baseline.

We additionally compare TimeInf to four commonly used anomaly detectors, covering classic approaches like Isolation Forest (Liu et al., 2008), a Long Short-Term Memory (LSTM)-based detector (Hundman et al., 2018), AR integrated moving average model (ARIMA) and vector AR model (VAR), and state-of-the-art transformer-based methods including Anomaly Transformer (Xu et al., 2021), TranAD (Tuli et al., 2022) and DCdetector (Yang et al., 2023). For AR-based models, we use ARIMA for univariate time series, and VAR for multivariate time series.

**Evaluation.** We apply Isolation Forest and LWCV directly to score each time point. Since the other methods assume an AR model, we use time blocks of length 100, as empirical results show optimal model performance at this length with minimal gains beyond; see Appendix Section D. TimeInf, block LOOCV, and conditional influence provide per-time-point scores directly. For LSTM, ARIMA/VAR, and Anomaly Transformer, we average scores across time blocks containing the target point to obtain its anomaly score.

Following the practice (Jiang et al., 2023), we apply the $k$-Means clustering algorithm to the calculated anomaly scores in our method, using a cluster of two to partition the scores into clusters of normal and anomaly time points. For other baselines, we consider two sets of anomaly predictions: (i) the $k$-Means clustering-based method used for TimeInf and (ii) the top-$k$ anomaly score-based method, where $k$ is the true number of anomalies. When computing performance metrics, we select the highest score between these two classification approaches. In practice, the exact number of anomalies is unknown, making this scoring procedure favorable to the other baselines. Evaluation metrics include F1 and AUC for detecting accuracy, along with the computation time cost for each method. We calculate the AUC by comparing the ground truth anomaly labels (0 for normal and 1 for anomalous data) with the anomaly scores produced by each baseline.

**Main Results.** As shown in Table 2, our method outperforms the other baselines in the UCR, SMAP, MSL, and NAB-Traffic datasets for anomaly detection. On the SMD dataset, our method achieves higher AUC than other methods. Notably, TimeInf outperforms other methods significantly on the UCR dataset, achieving a 0.79 AUC compared to the sub-0.65 AUC of its competitors (a 20% improvement). Compared to state-of-the-art transformer-based approaches, TimeInf outperforms TranAD by 7-30% and DCdetector by 3-17% in AUC across the datasets. Although we maintain a consistent block length of 100 across all datasets for a fair comparison, using a block length optimized for each dataset could potentially yield even better results, as shown by the ablation study in the Appendix (Figure 5). Additionally, TimeInf requires substantially less computation time than most baseline methods except Isolation Forest and conditional influence, enabling efficient data contribution assessment for large datasets.

## 4.2 INTERPRETABILITY OF TIMEINF

We investigate reasons behind the performance of TimeInf with a qualitative analysis using the UCR dataset (Wu & Keogh, 2021). Figure 1 compares the anomaly scores of TimeInf with other baselines across various types of time series anomalies, namely point anomaly, noisy data, local contextual,

and global contextual anomalies. Our method better captures challenging local and global contextual anomalies, whereas other baselines tend to identify only large deviations in data values, struggling to distinguish abrupt changes in a local or a global context. For point anomalies and noisy data, while other baselines can detect them, our method produces very low anomaly scores for the normal points, effectively distinguishing them from anomalies.

TimeInf outperforms Isolation Forest due to its consideration of temporal structure, allowing it to better handle contextual anomaly patterns. Unlike deep learning-based approaches such as LSTM and Anomaly Transformer, our method does not rely on training on an anomaly-free subset before being applied to detect anomalies in the contaminated subset. Due to this difference, TimeInf generally performs better than other prediction-based methods in identifying anomalies within the training dataset.

Among time series data contribution methods, Block LOOCV is a strong competitor to TimeInf. It directly calculates the effect of removing all time blocks containing the time point of interest by retraining the model following each data removal. However, Block LOOCV has several limitations. Calculating leave-one-out values requires computationally expensive model retraining for each time point, as influence functions cannot be used to approximate this process. It is also less sample-efficient and potentially less accurate. For instance, with 1000 training time blocks and a window size of 100, removing one time point eliminates 10% of the training data. We observe that TimeInf outperforms Block LOOCV in anomaly detection, as Block LOOCV's removal of large portions of training data compromises parameter estimation accuracy. Figure 1 reveals comparable patterns in the data values of TimeInf and Block LOOCV, as both methods aim to capture similar quantities. However, TimeInf is more efficient by approximating the inefficient model retraining procedure used in Block LOOCV.

TimeInf surpasses conditional influence in anomaly detection by considering multiple temporal arrangements from various time blocks containing the target point, rather than focusing on a single specific arrangement of preceding points. This approach enables TimeInf to identify anomalies across diverse contexts. TimeInf outperforms LWCV, likely because LWCV is constrained to latent state-space models, specifically an HMM with Gaussian emissions. This imposes a restrictive Markov assumption, considering only the immediate past. In contrast, TimeInf employs an AR model of order 100, allowing it to account for a much longer historical context. Figure 1 illustrates the superiority of TimeInf over conditional influence and LWCV. TimeInf produces less noisy data values by considering a broader range of temporal contexts in which the target time point may appear.

We have added more visualizations in Figures 10 and 11 in the Appendix to further compare the performance of different anomaly detectors to TimeInf. Moreover, we have explored other datasets, which can be found in Appendix Section F.

## 4.3 IDENTIFYING MISLABELED ANNOTATIONS IN ANOMALY DETECTION

Recent research by Wu & Keogh (2021) has exposed significant flaws in many anomaly detection datasets, casting doubt on the reliability of numerous published comparisons of anomaly detection algorithms. These benchmark datasets frequently contain mislabeled ground truth annotations. For example, Figures 2 and 3 illustrate two types of problematic annotations: segments with distinct anomalous patterns that lack anomaly labels, and segments labeled as anomalies that appear normal. Additional instances of mislabeling can be found in Figure 12 in the Appendix, as well as in Wu & Keogh (2021).

While our method excels in many real-world datasets, Appendix Table 4 shows that TimeInf is only comparable or slightly poor compared to Isolation Forest on problematic datasets such as PSM (Abdulaal et al., 2021), NAB-Tweets (Ahmad et al., 2017), SWaT (Mathur & Tippenhauer, 2016), and KDD-Cup99 (Stolfo et al., 2000). However, we believe that this poor performance is due to the low-quality problem of datasets, rather than the methods themselves. In fact, TimeInf provides highly intuitive scores in identifying real anomalies, rather than adhering to potentially mislabeled annotations, in many qualitative analyses. For instance, in Figure 2 Segment 3, TimeInf assigns higher anomaly scores to time points within segments not officially marked as anomalies, raising questions about the reliability of these labels. Figures 2 and 3 provide more examples of problematic ground truth annotations, showing how TimeInf can expose potential mislabeling in data, while Isolation Forest adheres to mislabeled annotations. This highlights a novel application of TimeInf to debugging and improving the quality of anomaly detection datasets.

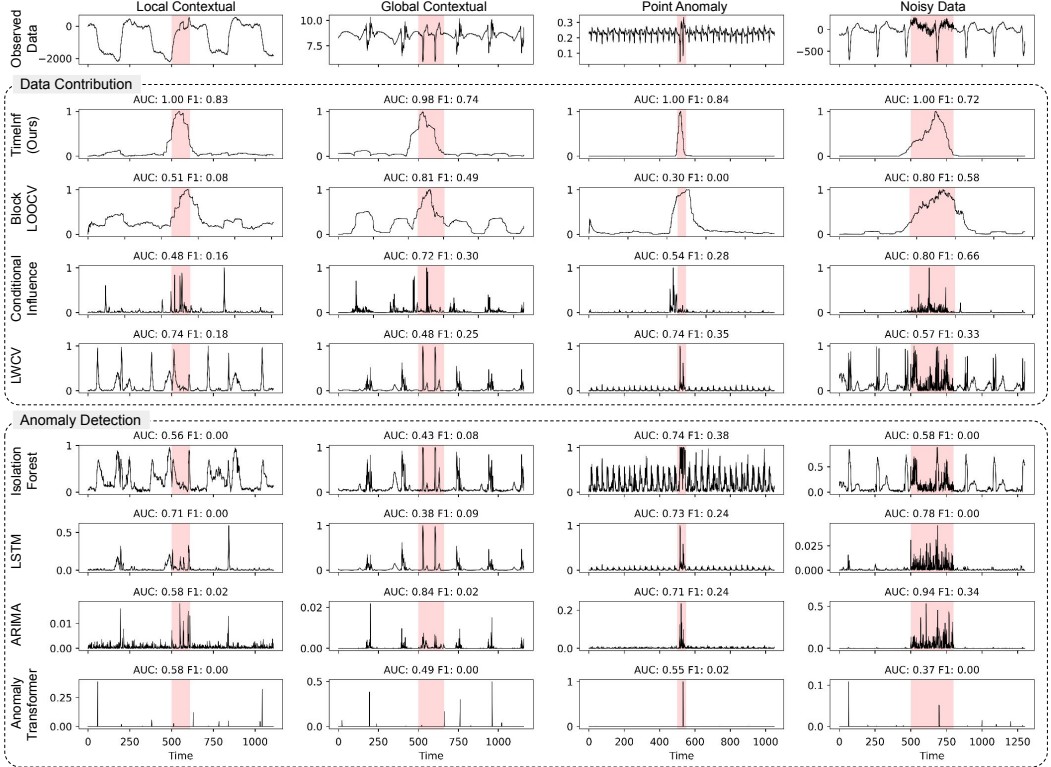

Figure 1: *Qualitative example of TimeInf and baseline methods in the UCR dataset.* The first row displays the observed time series, while the other rows show the anomaly scores generated by each method. For better visualizations, we normalize the anomaly scores from each method to a range between 0 and 1. The ground truth anomaly intervals are marked in red segments. TimeInf outperforms state-of-the-art time series anomaly detection methods and other data contribution techniques in identifying diverse anomaly patterns in terms of AUC and F1, and it also provides interpretable attributions.

## 5 CONCLUDING REMARKS

We propose TimeInf, an efficient method for measuring the contribution of time series data to model predictions. This approach serves as a unified framework, addressing limitations of existing time series data contribution methods. Our experiments demonstrate that TimeInf effectively identifies harmful time points through anomaly detection and provides intuitive data values, enabling easier distinction of diverse anomaly patterns. Moreover, TimeInf can expose mislabeled ground truth annotations in anomaly detection datasets, potentially improving training data quality. Its applications extend beyond anomaly detection; in Appendix Section E, we illustrate TimeInf's ability to identify helpful time points for time series forecasting through a data pruning experiment. Our research also shows that TimeInf is model-agnostic and robust against hyperparameter changes, as detailed in Appendix Sections C and D.

**Limitations.** Although TimeInf can be used to measure the influence of a data point on various metrics, this work focuses on the validation loss function. This choice highlights TimeInf's capability in time series anomaly detection and its potential applications (*e.g.*, identifying low-quality datasets in Section 4.3). We believe there are many promising applications for influence functions—detecting the most representative patterns under distribution shifts—but we leave this as an intriguing direction for future work. Another important challenge is that many data attribution methods, including TimeInf, often show unsatisfactory downstream task performance, especially when data are high-dimensional. We find that TimeInf's performance gains over other data contribution methods are marginal for high-dimensional data, which poses the question of whether this is due to low-quality data or methodological differences. We believe it requires a thorough investigation using high-quality multivariate time series datasets with reliable annotations.

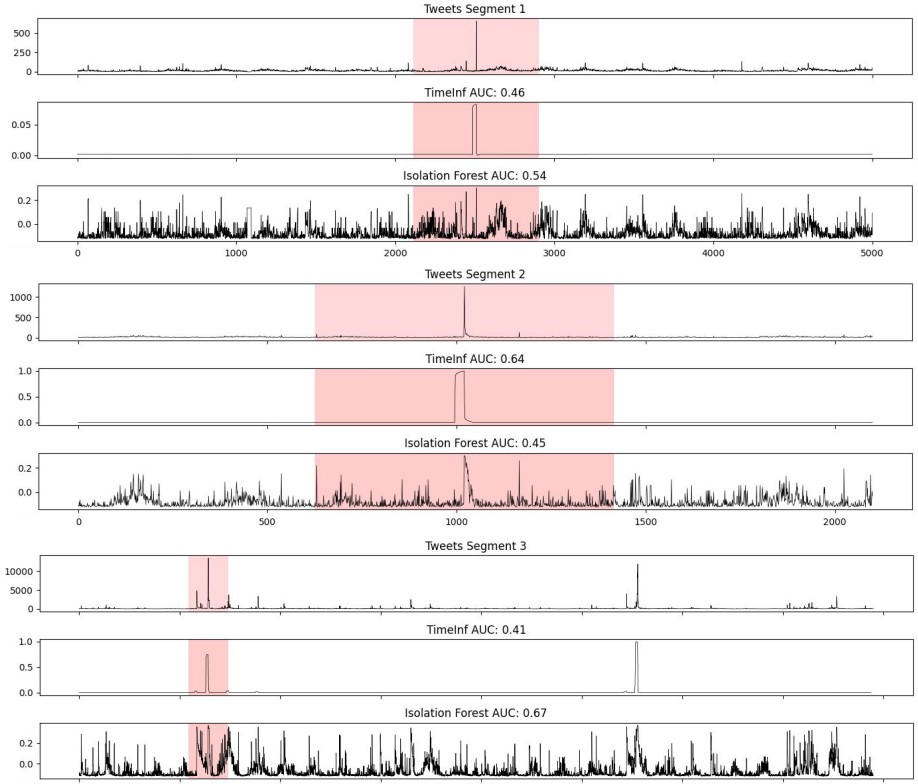

Figure 2: *A qualitative example of mislabeled anomaly points in the NAB-Tweets dataset.* Each panel displays a segment from the observed data in the first row, with anomaly scores from TimeInf and Isolation Forest in the following rows. Anomaly scores are normalized to a 0–1 range for clarity. The ground truth anomaly is marked in red. In NAB-Tweets segments 1 and 2, a large anomalous section is incorrectly labeled; most of it is normal, with only a small anomalous segment. Segment 3 shows a clear anomaly between time points 5000 and 6000 but lacks an anomaly label. Isolation Forest adheres to mislabeled annotations and obtains a high AUC, while TimeInf accurately identifies the anomalies but receives a low AUC.

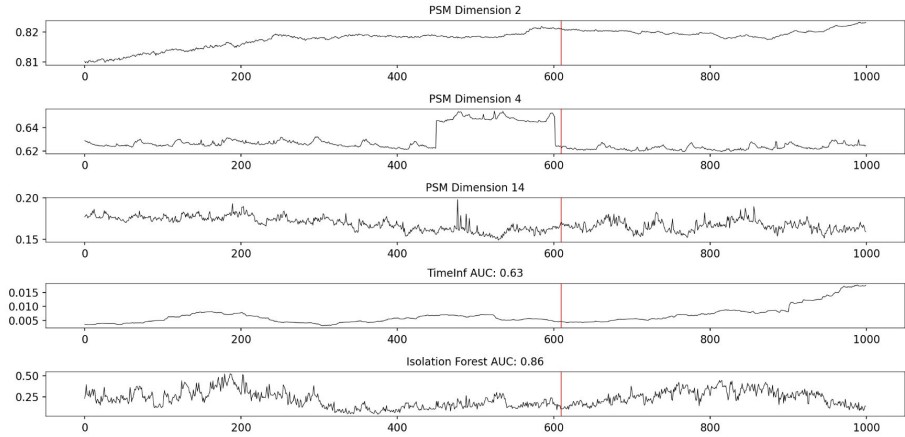

Figure 3: *A qualitative example of mislabeled anomaly points in the PSM dataset.* The first three rows display selected dimensions of the time series, while the remaining rows show normalized anomaly scores from TimeInf and Isolation Forest (range 0–1). The ground truth anomaly is marked in red. Although a point anomaly is labeled near time point 600, it appears normal, while the preceding segment seems more suspicious. TimeInf does not flag the labeled point as anomalous. Although TimeInf and Isolation Forest have similar anomaly scores near the labeled anomaly, Isolation Forest obtains a higher AUC, highlighting the problematic annotations.

## ACKNOWLEDGEMENTS

We acknowledge computing resources from Columbia University's Shared Research Computing Facility project, which is supported by NIH Research Facility Improvement Grant 1G20RR030893-01, and associated funds from the New York State Empire State Development, Division of Science Technology and Innovation (NYSTAR) Contract C090171, both awarded April 15, 2010.

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

## APPENDIX

## A. DERIVING INFLUENCE FUNCTIONS FOR M-ESTIMATORS

Let $\{X_t\}$ denote the random process. Denote $F^m$ the distribution of $X_t^{[m]} = (X_{t_1}, \ldots, X_{t_m})$. An M-estimator $\hat{\theta}$ for model parameters $\theta$ is the solution to a minimization problem over the data:

$$\hat{\theta} = \underset{\theta \in \Theta}{\operatorname{argmin}} \, \mathbb{E}[\rho(X_t^{[m]}, \theta)].$$

We consider a class of M-estimators for which $\rho$ is differentiable and convex (Planiden & Wang, 2014; Bertsekas et al., 2003), enabling us to express the estimate as the solution to a set of equations:

$$\mathbb{E}[\psi(X_t^{[m]}, \theta)] = 0, \quad \psi(X_t^{[m]}, \theta) = \frac{d\rho(X_t^{[m]}, \theta)}{d\theta}.$$

Influence functions compute the infinitesimal change in the estimator if the original distribution $F^m$ is contaminated by $z^{[m]}$ from $G^m = \delta_{z^{[m]}}$ by some small $\epsilon$ amount, giving us a contaminated estimator $\hat{\theta}_\epsilon = \underset{\theta_\epsilon \in \Theta}{\operatorname{argmin}} \, \mathbb{E}[\rho(X_t^{[m]}, z^{[m]}, \theta_\epsilon)]$. For the contaminated distribution $F_\epsilon^m$, $\hat{\theta}_\epsilon$ must satisfy:

$$\mathbb{E}[(1-\epsilon)\psi(X_t^{[m]}, \hat{\theta}_\epsilon) + \epsilon\psi(z^{[m]}, \hat{\theta}_\epsilon)] = (1-\epsilon)\mathbb{E}[\psi(X_t^{[m]}, \hat{\theta}_\epsilon)] + \epsilon\mathbb{E}[\psi(z^{[m]}, \hat{\theta}_\epsilon)] = 0.$$

We can differentiate to determine the effect of changing $\epsilon$ on $\hat{\theta}$:

$$\frac{d}{d\epsilon}(1-\epsilon)\mathbb{E}[\psi(X_t^{[m]},\hat{\theta}_\epsilon)] = -\frac{d}{d\epsilon}\epsilon\mathbb{E}[\psi(z^{[m]},\hat{\theta}_\epsilon)]$$

$$-\mathbb{E}[\psi(X_t^{[m]},\hat{\theta}_\epsilon)] + (1-\epsilon)\mathbb{E}[\frac{d\psi(X_t^{[m]},\hat{\theta}_\epsilon)}{d\theta}]\frac{d\hat{\theta}_\epsilon}{d\epsilon} = -\mathbb{E}[\psi(z^{[m]},\hat{\theta}_\epsilon)] - \epsilon\mathbb{E}[\frac{d\psi(z^{[m]},\hat{\theta}_\epsilon)}{d\theta}]\frac{d\hat{\theta}_\epsilon}{d\epsilon}.$$

Around $\epsilon = 0$, the estimator must satisfy $\mathbb{E}[\psi(X_t^{[m]},\hat{\theta})] = 0$, thus the influence function for model parameters in Equation (3) is

$$\frac{d\hat{\theta}_\epsilon}{d\epsilon}\Big|_{\epsilon=0} = -\mathbb{E}[\frac{d\psi(X_t^{[m]},\hat{\theta})}{d\theta}]^{-1}\mathbb{E}[\psi(z^{[m]},\hat{\theta})].$$

Applying the chain rule, the effect of changing $\epsilon$ on the loss evaluated at a test point $z_{\text{test}}$ in Equation (4) is then

$$\frac{d\rho(z_{\text{test}},\hat{\theta}_\epsilon)}{d\epsilon}\Big|_{\epsilon=0} = \frac{d\rho(z_{\text{test}},\hat{\theta})}{d\theta}\frac{d\hat{\theta}_\epsilon}{d\epsilon}\Big|_{\epsilon=0} = -\psi(z_{\text{test}},\hat{\theta})\mathbb{E}[\frac{d\psi(X_t^{[m]},\hat{\theta})}{d\theta}]^{-1}\mathbb{E}[\psi(z^{[m]},\hat{\theta})].$$

---

**Algorithm 1** TimeInf Estimator

---

**Require:** Time point of interest $z$, training time blocks $\{x_1^{[m]},\ldots,x_n^{[m]}\}$, test time block $z_{\text{test}}^{[m]}$, index set of time blocks that contain the time point of interest $S_z$, model parameters $\hat{\theta}$, loss function $l$, and block length $m$.

**Ensure:** TimeInf $\mathcal{I}_{\text{time}}(z, z_{\text{test}}^{[m]})$

  (Optional if $\hat{\theta}$ is ready) Learn the parameter $\hat{\theta}$ by training a model on $\{x_1^{[m]},\ldots,x_n^{[m]}\}$.

  1. Compute the gradients at the training time block $\psi(x_i^{[m]},\hat{\theta}) = \frac{d}{d\theta}l(x_i^{[m]},\hat{\theta})$ for all $i \in \{1,\ldots,n\}$ and the Hessian $H_{\hat{\theta}}$.

  2. Compute the gradient at the test time block $\psi(z_{\text{test}}^{[m]},\hat{\theta}) = \frac{d}{d\theta}l(z_{\text{test}}^{[m]},\hat{\theta})$

  **for** $i \in S_z$ **do**

   $\mathcal{I}_{\text{time}}(z, z_{\text{test}}^{[m]}) = \frac{1}{|S_z|}\left(-\psi(z_{\text{test}}^{[m]},\hat{\theta})^\top H_{\hat{\theta}}^{-1}\psi(x_i^{[m]},\hat{\theta}) + \mathcal{I}_{\text{time}}(z, z_{\text{test}}^{[m]})\right).$

  **end for**

---

## B. BLACK-BOX MODELS AND NON-DIFFERENTIABLE LOSSES

Although we focus on linear AR models in the main paper due to their computational efficiency, the computation of TimeInf is model-agnostic, allowing us to explain any models of interest under our proposed framework. For "black-box" models with differentiable losses, like RNN and LSTM, we compute the Hessian $\mathbb{E}[\frac{d\psi(X_t^{[m]},\hat{\theta})}{d\theta}]$ in Equation (4) using automatic differentiation packages. The primary challenge lies in computing the inverse Hessian-vector products, and various solutions proposed by Koh & Liang (2017), Grosse et al. (2023), Kwon et al. (2023), Kim et al. (2023), Pruthi et al. (2020), and others have scaled this computation to deep neural networks. For linear AR models, we simply compute the inverse of the Hessian. However, for black-box models, this computation becomes intractable. Therefore, we employ the CG method in Koh & Liang (2017) and TracIn (Pruthi et al., 2020) to approximate the inverse of the Hessian.

For non-differentiable models such as tree-based models and ensemble models, we compute non-parametric influence functions (Feldman & Zhang, 2020). The idea is to calculate a Leave-One-Out (LOO) data value by excluding a data point from the training set, retraining the model, and evaluating the impact of this removal on prediction performance. Given its computational costs, Feldman & Zhang (2020) employs data subsampling to approximate LOO. The nonparametric counterpart of $\mathcal{I}_{\text{block}}$ in Equation (4) is:

$$\mathcal{I}_{\text{nonparametric}}(z^{[m]}, z_{\text{test}}^{[m]}) = \mathbb{E}_{k\sim[K]}[\rho(z_{\text{test}}^{[m]},\hat{\theta}_{S_k}) \mid z^{[m]} \in S_k] - \mathbb{E}_{k\sim[K]}[\rho(z_{\text{test}}^{[m]},\hat{\theta}_{S_k}) \mid z^{[m]} \notin S_k],$$

where $\rho$ denotes the loss function, $\hat{\theta}$ represents the learned parameters of the model under consideration. The notation $\hat{\theta}_{S_k}$ is used for the parameters of the model fitted on a subset $S_k$. The process

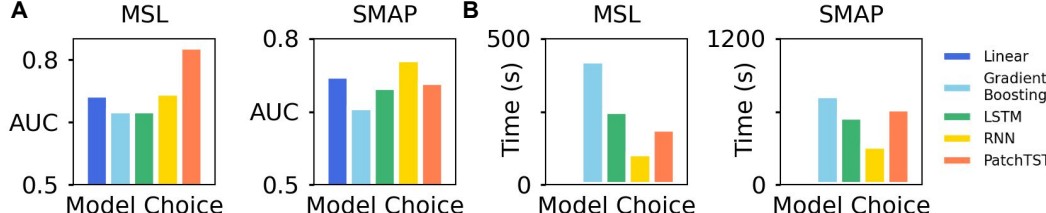

Figure 4: *The performance and computation time of TimeInf across different model choices on the MSL and SMAP datasets.* Panels (A) and (B) show the mean AUC and mean runtime in seconds for each model choice. Linear AR models provide the fastest computation for TimeInf, with negligible time costs not visible in panel (B) due to their small computation time. Choosing a more complex model improves the performance of TimeInf but at the expense of increased computation time.

involves $K$ subsets denoted as $S_k$, which are randomly sampled from the training data of size $n$ with a smaller sample size $n_{\text{subset}} \ll n$. Subsequently, the models of interest are trained separately on each sampled subset. Once equipped with $\mathcal{I}_{\text{nonparametric}}$, we can compute the influence of a time point ($\mathcal{I}_{\text{time}}$), self-influence ($\mathcal{I}_{\text{self}}$), and the influence of a time point on the test prediction ($\mathcal{I}_{\text{test}}$), according to Equations 6, 8 and 9.

## C. MODEL AGNOSTICITY

Although we used linear AR models to compute TimeInf for anomaly detection, TimeInf is model-agnostic; see details in Appendix Section B. To understand the variation in performance and computation time across different models, we compare TimeInf computed using linear AR models against an RNN (1,153 params), LSTM (4,513 params), transformer-based PatchTST (Nie et al., 2022) (299,073 params), and a gradient boosting regressor with 100 trees of max depth 3. All models use MSE as the loss function. For RNN and LSTM, we apply the CG used in Koh & Liang (2017) to compute TimeInf for differentiable black-box models. For the large nonlinear PatchTST model, we use the scalable "Hessian-free" (Pruthi et al., 2020) method to compute TimeInf. For gradient boosting, we compute nonparametric TimeInf according to Feldman & Zhang (2020); see details in Appendix Section B. To ensure fair comparisons, all models are thoroughly trained and perform well on held-out validation sets.

We evaluate TimeInf computed with different model choices on the SMAP and MSL datasets, as shown in Figure 4. Panel (A) shows that the performance of TimeInf in anomaly detection improves with the transition from a linear AR model to nonlinear models such as LSTM, RNN and PatchTST. In Panel (B), there is a slight increase in computation time when a nonlinear model is used. The computation of nonparametric TimeInf is time-consuming due to the need for data subsampling and model re-training.

In summary, using higher dimensional or more complex models improves TimeInf performance but increases computation time. The choice between a complex model for better performance or a simple linear AR for efficiency depends on the user's needs. Moreover, these findings suggest that the results in Table 2 could be further improved using flexible deep neural network models. We highlight that the computation time for TimeInf is small, even for moderately-sized transformers (300K parameters). Therefore, TimeInf can potentially scale to large over-parameterized models, enabling interpretability of time series foundation models. See the ablation study in Appendix Section D (Figure 6) for detailed performance comparisons across model sizes.

## D. ABLATION STUDY

This section explores how hyperparameters of TimeInf – the block length used for constructing the empirical contaminating distribution $\hat{G}^m$ in Equation (6), and the model size – affect the downstream task performance through an ablation study.

**Experiments on Different Block Lengths.** The performance of TimeInf in anomaly detection can be affected by the block length, which determines the order of the AR model used in its computation.

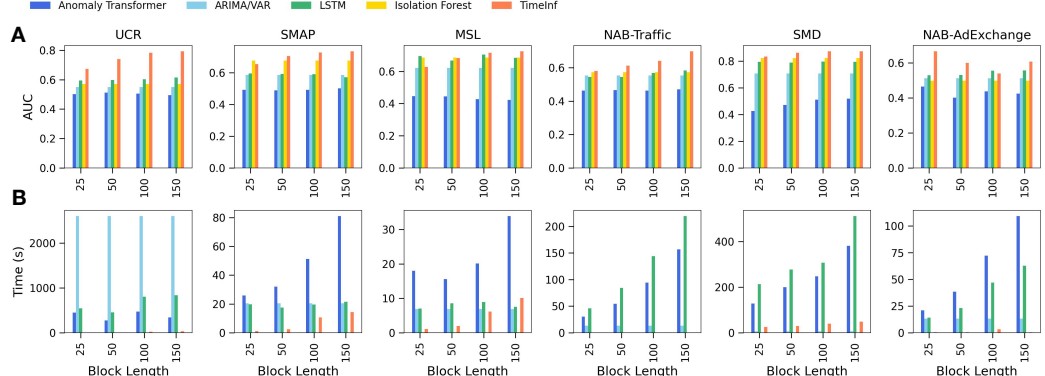

Figure 5: *The performance and computation time of TimeInf and baseline methods across different block lengths.* Panels (A) and (B) show the mean AUC and mean runtime in seconds at the block lengths of 25, 50, 100, and 150. Compared to other methods, TimeInf achieves a higher AUC while requiring significantly lower computation time across various datasets.

Table 3: *The performance of TimeInf across different time strides.* Ablation studies on the SMAP and SMD datasets show that the anomaly detection AUC decreases as the time stride increases.

| Time Stride | 1 | 5 | 10 | 20 | 50 |
|---|---|---|---|---|---|
| SMAP | 0.73 | 0.64 | 0.65 | 0.61 | 0.58 |
| SMD | 0.87 | 0.88 | 0.88 | 0.83 | 0.70 |

Figure 5 shows how AUC and computation time vary across block lengths of 25, 50, 100, and 150 for each anomaly detector. In Figure 5 (A), we observe that the AUC of TimeInf improves when block length becomes larger across most datasets, as larger block length enables TimeInf to better capture contextual anomalies that persist for an extended duration. Panel (B) indicates an increase in computation time when block length increases for most methods. This is because an extended block length increases the dimension of the model parameters, and thus increases the time needed for computing the inverse Hessian. In summary, TimeInf consistently outperforms other anomaly detectors across a wide range of block lengths, while requiring significantly less time.

**Experiments on Time Stride.** The time stride, which determines the number of time blocks a time point appears in, affects the anomaly detection performance of TimeInf. Table 3 shows that as the time stride increases, the AUC decreases, with the threshold varying across datasets. Larger strides provide fewer time blocks containing the point, limiting the sampling of temporal configurations around it. Consequently, with fewer samples, TimeInf estimate becomes less reliable, which is why we choose stride 1 in our experiments.

**Experiments on Model Size.** Experiments are conducted to examine the effect of model size on TimeInf's performance. The anomaly detection results on the MSL and SMAP datasets using TimeInf computed with models of different sizes are compared. As depicted in Figure 6 panel (A), the anomaly detection performance of TimeInf improves with increasing model size on the MSL dataset, with the transformer-based PatchTST model scoring the highest. For the SMAP dataset, although PatchTST performs worse, the nonlinear RNN model achieves the best performance. Despite the computation time increasing with model size, PatchTST with over 300 thousand parameters remains fast, enabling scaling to larger deep learning models.

## E. DATA PRUNING

Identifying the most influential time patterns in the training dataset can improve forecasting performance by prioritizing patterns that contribute significantly to accurate predictions. In this section, we

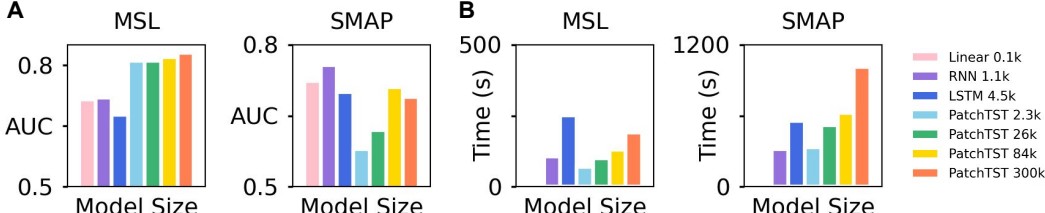

Figure 6: *The performance and computation time of TimeInf across different model sizes on the SMAP and MSL datasets.* Panels (A) and (B) show the mean AUC and mean runtime in seconds for each model size. Linear AR models are the smallest and provide the fastest TimeInf computation, with negligible time costs not visible in panel (B) due to their small computation time. Choosing a larger model improves the performance of TimeInf but at the expense of increased computation time.

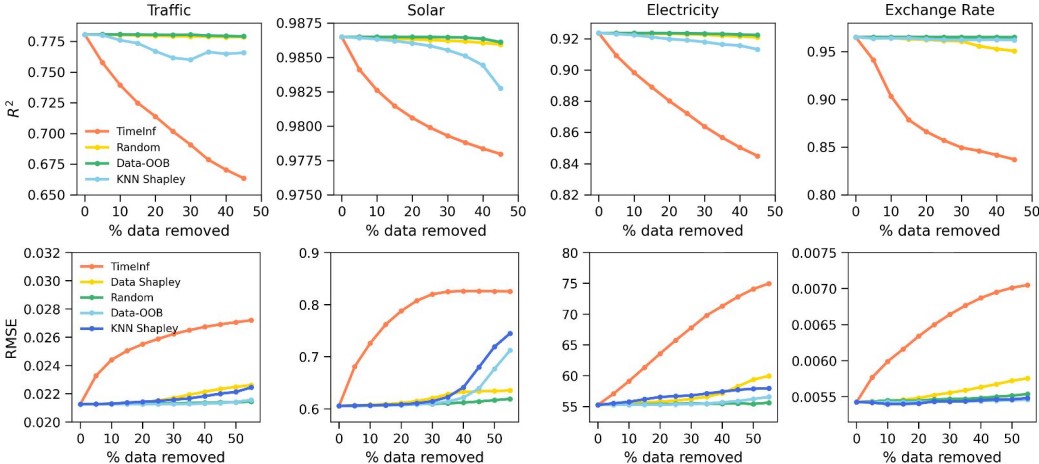

Figure 7: *Data pruning results for TimeInf and other data contribution methods.* We estimate the contribution of each time block in the training data using various baselines. For each method, we iteratively remove the most influential time block from the training set and track the resulting performance degradation, measured by $R^2$ and RMSE. The better the data contribution method, the lower the $R^2$ curve and the higher the RMSE curve. TimeInf shows a steeper decrease in $R^2$ and steeper increase in RMSE than state-of-the-art data contribution methods, demonstrating superior performance in identifying helpful data patterns for time series forecasting across four real-world datasets.

demonstrate TimeInf's ability to identify helpful time patterns for forecasting through a data pruning experiment.

**Experimental Settings.** For each dataset, we conduct 100 independent experiments, each time randomly sampling 5,000 consecutive data points. These points are then sequentially partitioned into a training set, a validation set, and a test set, with fixed sizes of 3000, 1000, and 1000 points, respectively.

Contrary to the point removal benchmark commonly used in data valuations (Jiang et al., 2023), our experiment employs a *block-wise* data pruning approach, which is more in line with common practices in time series forecasting. We divide the time series into equally sized blocks. Each block within the training dataset is assigned a value indicating its contribution to future time series forecasting. Subsequently, the blocks are removed in descending order of the values. After each block is removed, we train a linear model on the remaining dataset and assess its performance on a held-out test set.

**Datasets.** We consider four datasets commonly used in time series forecasting literature (Lai et al., 2018): Traffic, Solar Energy, Electricity (Trindade, 2015), and Exchange Rate (Lai et al., 2018). The Traffic dataset[4] aggregates hourly road occupancy rates over 48 months from 862 sensors across California. The Solar Energy dataset[5] comprises records from 137 photovoltaic power plants in

---

[4]https://pems.dot.ca.gov/
[5]https://www.nrel.gov/grid/solar-power-data.html

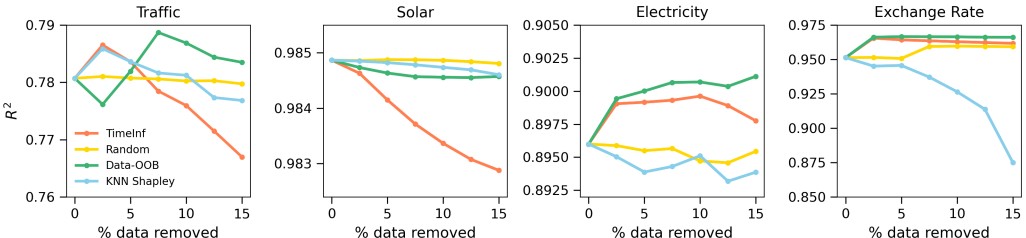

Figure 8: *Quantitative results for TimeInf and other data contribution methods from a data pruning experiment across four real-world datasets.* We estimate the contribution of each time block in the training data using various baselines. For each method, we iteratively remove the least helpful time block from the training set and track the resulting performance degradation, measured by $R^2$, aiming for a less rapid decrease or even an increase in performance.

Alabama State. The Electricity dataset includes the electricity consumption patterns of 321 clients from 2012 to 2014 (Trindade, 2015). The Exchange Rate dataset includes daily exchange rates for 8 countries, covering the period from 1990 to 2016 (Lai et al., 2018).

**Baselines.** For each block in the training dataset, we assess its impact by averaging the score $\mathcal{I}_{\text{block}}$ from Equation (4) across all blocks in the validation set. We fix the block length as 100. For anomaly detection, standard methods are available, but for data pruning, no established method exists. Therefore, we employ the state-of-the-art data valuation methods as baselines: Shapley-value-based data valuation methods, namely KNN Shapley (Jia et al., 2019), and out-of-bag estimates of data contribution, such as Data-OOB (Kwon & Zou, 2023). We exclude Data Shapley (Ghorbani & Zou, 2019) from consideration due to its computational inefficiency, making it difficult to apply to our datasets.

For a fair comparison, we consistently employ a linear AR model across all baseline data valuation methods. We assign a data value to each time block in the training dataset and iteratively remove time blocks, starting from the most helpful to the least helpful for forecasting on the validation set. After each removal, we train a linear AR model on the remaining dataset and assess the model predictions on the held-out test set.

**Evaluation.** We use $R^2$ and root mean squared error (RMSE) as evaluation metrics for model predictions. A larger $R^2$ and smaller RMSE correspond to a better prediction performance.

**Main Results.** Figure 7 illustrates the effectiveness of TimeInf in identifying the most valuable patterns for forecasting in four real-world datasets. We found that predictive performance declines more rapidly with TimeInf, suggesting its superior efficacy compared to other baseline methods. For instance, in the Traffic dataset, we observe a more than 10% drop in $R^2$ after removing 30% of the blocks, whereas other methods result in less than a 2% decrease. Figure 9 illustrates a qualitative example from the Electricity, which demonstrates how TimeInf successfully identifies influential periodic patterns for test forecasting.

Additionally, TimeInf is compared to baseline methods in identifying the least helpful (or most harmful) patterns for forecasting. Time blocks are iteratively removed from least helpful to most helpful based on data values assigned by each method. After each removal, a linear AR model is trained and evaluated. TimeInf can correctly identify the least helpful temporal patterns in most datasets except the Solar dataset, as shown in Figure 8.

## F. ANOMALY DETECTION EXPERIMENT DETAILS

**Dataset Details.** In the main paper, we extensively evaluate TimeInf on five benchmark datasets. The univariate datasets include UCR (Wu & Keogh, 2021), featuring 250 time series from diverse fields, and NAB-Traffic, which contains real-time traffic data from the Twin Cities Metro area. For multivariate datasets, MSL (Mars Science Laboratory) and SMAP (Soil Moisture Active Passive satellite) are NASA datasets (Hundman et al., 2018) monitoring spacecraft telemetry anomalies. SMD (Server Machine Dataset) is a dataset from an Internet company (Su et al., 2019).

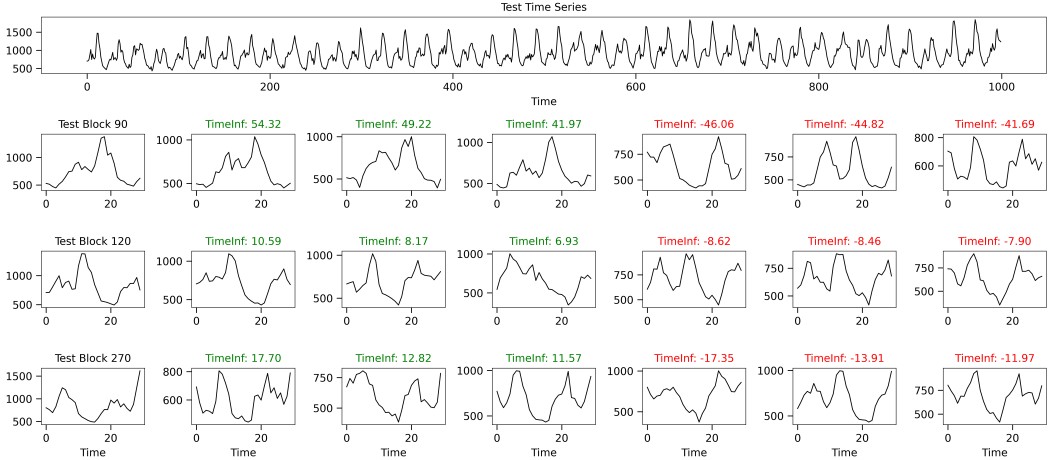

Figure 9: *Qualitative results of the data pruning experiment for TimeInf on the Electricity dataset.* The first row shows the raw test time series, with subsequent rows depicting the most influential (green) and detrimental (red) time blocks in the training data linked to the example test block (black) based on TimeInf. TimeInf effectively captures the periodic pattern in the time series, offering valuable insights for forecasting.

Table 4: *Quantitative results for TimeInf and other anomaly detectors across seven real-world datasets.* For both F1 and AUC metrics, a higher value indicates a better performance. TimeInf faces challenges in these datasets due to issues with the reliability of ground truth anomaly labels; see Appendix Section F for details.

|  | NAB-Tweets | | NAB-Taxi | | PSM | | SWaT | | WADI | | KDD-Cup99 | | NAB-Exchange | |
|---|---|---|---|---|---|---|---|---|---|---|---|---|---|---|
|  | AUC | F1 | AUC | F1 | AUC | F1 | AUC | F1 | AUC | F1 | AUC | F1 | AUC | F1 |
| Isolation Forest | 0.55 | 0.18 | 0.64 | 0.13 | 0.71 | 0.51 | 0.87 | 0.69 | 0.74 | 0.34 | 0.74 | 0.41 | 0.50 | 0.11 |
| LSTM | 0.55 | 0.17 | 0.33 | 0.04 | 0.62 | 0.38 | 0.30 | 0.16 | 0.49 | 0.08 | 0.98 | 0.88 | 0.56 | 0.12 |
| ARIMA / VAR | 0.56 | 0.17 | 0.49 | 0.05 | 0.55 | 0.33 | 0.46 | 0.11 | 0.41 | 0.04 | 0.87 | 0.52 | 0.51 | 0.14 |
| Anomaly Transformer | 0.50 | 0.08 | 0.51 | 0.06 | 0.48 | 0.15 | 0.60 | 0.22 | 0.50 | 0.06 | 0.54 | 0.17 | 0.44 | 0.07 |
| TimeInf (Ours) | 0.52 | 0.16 | 0.52 | 0.02 | 0.63 | 0.02 | 0.61 | 0.08 | 0.63 | 0.14 | 0.79 | 0.43 | 0.54 | 0.34 |

In Table 4, we include additional datasets: (i) NAB-Tweets (Ahmad et al., 2017) comprises Twitter mentions of publicly-traded companies such as Google and IBM. (ii) NAB-Taxi (Ahmad et al., 2017) records NYC taxi passenger numbers with anomalies during events like the NYC marathon and holidays. (iii) The Pooled Server Metrics (PSM) (Abdulaal et al., 2021) dataset from eBay captures anomalies in application server nodes. (iv) The Secure Water Treatment (SWaT) (Mathur & Tippenhauer, 2016) dataset features data collected from a sewage water treatment facility, where the anomalies are caused due to cyberattacks. (v) The Water Distribution (WADI) (Ahmed et al., 2017) dataset is a distribution system consisting of a large number of water distribution pipelines. (vi) KDD-Cup99 (Stolfo et al., 2000) contains network connection data for intrusion detection in a military network environment. (vii) NAB-AdExchange collects online advertisement clicking rates. Details of all datasets are summarized in Table 5. Although our method performs well in many real-world datasets, TimeInf faces challenges in these datasets due to issues with the reliability of ground truth anomaly labels. Visual examples of problematic anomaly annotations are in Figure 12, 3, and 2, illustrating how TimeInf can expose potential mislabeling in the data.

The UCR, SMAP, MSL, SMD, PSM, and SWaT datasets have a pre-partitioned structure: a clean training set without anomalies and a contaminated test set with anomalies. In these datasets, we exclusively train and evaluate the baseline anomaly detection methods on the contaminated test set. Conversely, WADI, KDD-Cupp99, NAB-Traffic, NAB-Tweets, and NAB-Taxi have a single contaminated dataset with dispersed anomalies. For these datasets, baseline methods are trained and evaluated on the entire dataset.

**Baseline Details.**    For block LOOCV, we employ a time block length of 100 and utilize a linear AR model for prediction. The conditional influence Kunsch (1984) method is implemented following the approach of Grosse et al. (2023), using a linear AR model with an order of 100. For LWCV, we

Table 5: *Details of the anomaly detection datasets.* Note that all discrete-valued dimensions are excluded from each dataset. The term "Average Length" refers to the average length across all time series in the dataset.

| Dataset | Dimensions | Num. of Time Series | Average Length | Average Anomaly Ratio |
|---|---|---|---|---|
| UCR | 1 | 250 | 56,205 | 0.8% |
| SMAP | 25 | 55 | 8,068 | 12.8% |
| MSL | 55 | 27 | 2,730 | 10.5% |
| NAB-Traffic | 1 | 7 | 2238 | 10.0% |
| SMD | 38 | 28 | 25,300 | 4.2% |
| NAB-AdExchange | 1 | 6 | 1602 | 10.0% |
| NAB-Tweets | 1 | 10 | 15863 | 9.9% |
| NAB-Taxi | 1 | 1 | 10320 | 5.2% |
| PSM | 25 | 1 | 87841 | 27.8% |
| SWaT | 51 | 1 | 449919 | 12.1% |
| WADI | 123 | 1 | 172,751 | 5.7% |
| KDD-CUP99 | 42 | 1 | 494,021 | 19.7% |

employ a HMM with Gaussian observations (Ghosh et al., 2020), incorporating two discrete states: one hidden state representing normal conditions, and another representing abnormal conditions.

## G. APPLYING TIMEINF TO MULTIVARIATE TIME SERIES

For anomaly detection in multivariate time series, we first apply self-influence to each dimension individually, and then average the anomaly scores across all dimensions to derive the final anomaly scores for individual time points. We refer to this approach as "SepInf" for simplicity. We empirically find that "SepInf" demonstrates better performance than directly applying self-influence to the multivariate time series, which we refer to as "MultiInf". In this section, we explore potential challenges associated with using MultiInf and illustrate why it may not be suitable for anomaly detection.

Suppose we have $T$ training time blocks of size $m$, $\{x_t^{[m]}\}_{t=1}^T$, $x_t^{[m]} = (x_t, x_{t-1}, \ldots, x_{t-m+1}) \in \mathbb{R}^{m \times p}$, where $p$ is the dimension of the multivariate time series. To compute MultiInf, we fit a vector AR model, i.e., multivariate linear regression, as follows

$$x_{t+1} = x_t \theta + u + \epsilon_{t+1}, \tag{10}$$

where, $x_{t+1} \in \mathbb{R}^p$ represents the one-step ahead prediction, while $x_t^{[m]} \in \mathbb{R}^{m \times p}$ denotes the past time blocks used for prediction. Additionally, $u \in \mathbb{R}^p$ and $\epsilon_{t+1} \in \mathbb{R}^p$ correspond to the intercept and error terms, respectively. The parameters of the AR model, denoted as $\theta = \{\theta_j\}_{j=1}^p \in \mathbb{R}^{p \times m \times p}$, are capable of capturing correlations across each dimension of the multivariate time series, between timesteps within a time block, and between each timestep in a time block and each dimension. This is expressed as follows:

$$\theta_j = \begin{bmatrix} \theta_{j11} & \theta_{j12} & \ldots & \theta_{j1p} \\ \theta_{j21} & \theta_{j22} & \ldots & \theta_{j2p} \\ \vdots & \ldots & \ldots & \vdots \\ \theta_{jm1} & \theta_{jm2} & \ldots & \theta_{jmp} \end{bmatrix} \in \mathbb{R}^{m \times p} \tag{11}$$

There are two practical challenges associated with computing the influence functions and over-weighting a time block on the parameters $\theta$: (1) Anomalies may not occur simultaneously in all dimensions of the time series. Consequently, capturing correlations among feature dimensions could result in poor anomaly scores because the ground truth anomaly labels may depend only on anomalies in some dimensions but not the others; see Figures 3, 11, and 12 for examples. (2) There may not be enough training samples to accurately estimate the high-dimensional parameters $\theta$ of size $p \times m \times p$. Inaccurately estimated $\theta$ can lead to the computation of inaccurate influence functions. For instance, when fitting the model to a 100-dimensional time series with a block length of 100, we have to learn 1 million model parameters, demanding a large amount of training data. (3) Additionally, using MultiInf causes issues computationally because the inversion of the Hessian matrix scales with the number of model parameters.

In contrast, SepInf is computed by initially applying self-influence to each dimension separately and then averaging the anomaly scores (scaled self-influences) across all dimensions of the multivariate time series. This is equivalent to fitting a multiple linear regression in the same form as Equation (10) under the assumption that each dimension is independent. This requires learning $\{\theta_j\}_{j=1}^p$ with

$$\theta_j = [\theta_{j1} \quad \theta_{j2} \quad \ldots \quad \theta_{jm}] \in \mathbb{R}^m,$$

since the off-diagonal entries in Equation (11) are zero. This approach effectively reduces the number of parameters to learn. Furthermore, since anomaly patterns are often vastly different between dimensions, and the ground truth anomaly labels may be influenced by only some feature dimensions, computing SepInf effectively separates the influences of time points from different dimensions, thereby mitigating potential issues stemming from multicollinearity. Moreover, computing SepInf is more efficient because it involves fewer parameters when constructing the inverse Hessian matrix.

## H. USING TIMEINF FOR VARIOUS DOWNSTREAM TASKS

In the paper, we use anomaly detection and time series forecasting as representative examples to demonstrate that TimeInf can identify both harmful and helpful data points in the data. In this section, we outline the setup for time series forecasting and anomaly detection, as well as how to apply TimeInf to both tasks.

### H1. TIME SERIES FORECASTING

Let $\{x_t\}_{t=1}^T$ represent a univariate time series, with the objective of predicting future values based on past observations. We define an input window of size $m$ as $x_t^{[m]} = (x_t, \ldots, x_{t-m+1})$. Let $f_\theta$ be a forecasting model with parameters $\theta$, such as an autoregressive model or a neural network. The forecasted value at time $t+1$ is given by $f_\theta(x_t^{[m]})$.

To evaluate the model performance, we use a loss function $\ell(x_{t+1}, f_\theta(x_t^{[m]}))$, such as the mean squared error (MSE). For time series forecasting, we are particularly interested in understanding the impact of a training data point on model performance at test time. Therefore, we partition the original time series into training and test sets.

First, we train the model on the training data to obtain the parameters $\hat{\theta}$ and the Hessian matrix $H_{\hat{\theta}}$. We can then compute the influence of any time window $z_t^{[m]}$ in the training data on any time window $z_{\text{test},t}^{[m]}$ in the test data as follows:

$$\mathcal{I}_{\text{block}}(z_t^{[m]}, z_{\text{test},t}^{[m]}) = -\frac{d}{d\theta}\ell(z_{\text{test},t+1}, f_{\hat{\theta}}(z_{\text{test},t}^{[m]}))^\top H_{\hat{\theta}}^{-1} \frac{d}{d\theta}\ell(z_{t+1}, f_{\hat{\theta}}(z_t^{[m]})).$$

To measure the impact of a specific time point $z_t$ in the training data on the entire test set, we define the following quantity:

$$\mathcal{I}_{\text{test}}(z_t) = \frac{1}{|S_{\text{test}}|}\frac{1}{|S_{z_t}|}\sum_{z_{\text{test},t}^{[m]} \in S_{\text{test}}}\sum_{z_t^{[m]} \in S_{z_t}}\mathcal{I}_{\text{block}}(z_t^{[m]}, z_{\text{test},t}^{[m]}),$$

where $S_{z_t}$ is the set of all time windows in the training data that include the time point $z_t$, and $S_{\text{test}}$ is the set of all time windows in the test set. The quantity $\mathcal{I}_{\text{test}}(z_t)$ measures the influence of upweighting a training time point on the test loss.

The interpretation of the influence of this time point varies depending on the chosen loss function. For instance, if we use MSE as the loss function, a negative value of $\mathcal{I}_{\text{test}}(z_t)$ indicates that upweighting $z_t$ will lower the MSE, suggesting that $z$ is important for good forecasting performance. On the contrary, if $\mathcal{I}_{\text{test}}(z_t)$ is positive, upweighting $z_t$ will increase the MSE, implying that $z_t$ is detrimental to forecasting accuracy.

### H2. TIME SERIES ANOMALY DETECTION

For anomaly detection, the objective is to identify anomalies by reconstructing the time series and comparing the reconstructions with the actual observations. Let $g_\phi$ be a reconstruction model

with an input window of size $m$, defined as $x_t^{[m]} = (x_t, \ldots, x_{t-m+1})$. The reconstructed time window is given by $g_\phi(x_t^{[m]})$. The anomaly score is based on the reconstruction error, expressed as $\ell(x_t^{[m]}, g_\phi(x_t^{[m]}))$. The underlying assumption is that if a time point is anomalous, reconstructing it using its own values should be easier than using normal points, as normal time points show distinctly different behaviors from anomalous ones.

Thus, we focus on measuring the reconstruction error without needing to partition the time series into training and test sets; we treat the entire series as a single dataset. First, we train the model on this data to obtain the parameters $\hat{\phi}$ and the Hessian matrix $H_{\hat{\phi}}$. We can then compute the influence of any time window $z_t^{[m]}$ on itself as follows:

$$\mathcal{I}_{\text{block}}(z_t^{[m]}, z_t^{[m]}) = -\frac{d}{d\phi}\ell(z_t^{[m]}, g_{\hat{\phi}}(z_t^{[m]}))^\top H_{\hat{\phi}}^{-1}\frac{d}{d\phi}\ell(z_t^{[m]}, g_{\hat{\phi}}(z_t^{[m]})).$$

Next, we compute self-influence, which measures the impact of upweighting a time point on its own reconstruction loss, as our anomaly detection metric:

$$\mathcal{I}_{\text{self}}(z_t) = \frac{1}{|S_{z_t}|}\sum_{z_t^{[m]} \in S_{z_t}} \mathcal{I}_{\text{block}}(z_t^{[m]}, z_t^{[m]}).$$

Using MSE as the reconstruction error, a negative $\mathcal{I}_{\text{self}}(z_t)$ suggests an anomaly, as upweighting this point improves its own reconstruction. This is based on the rationale that reconstructing an anomaly is easier with abnormal points than with normal ones. We take the absolute value of self-influence to compute the anomaly scores, with higher scores indicating a higher likelihood of being an anomaly.

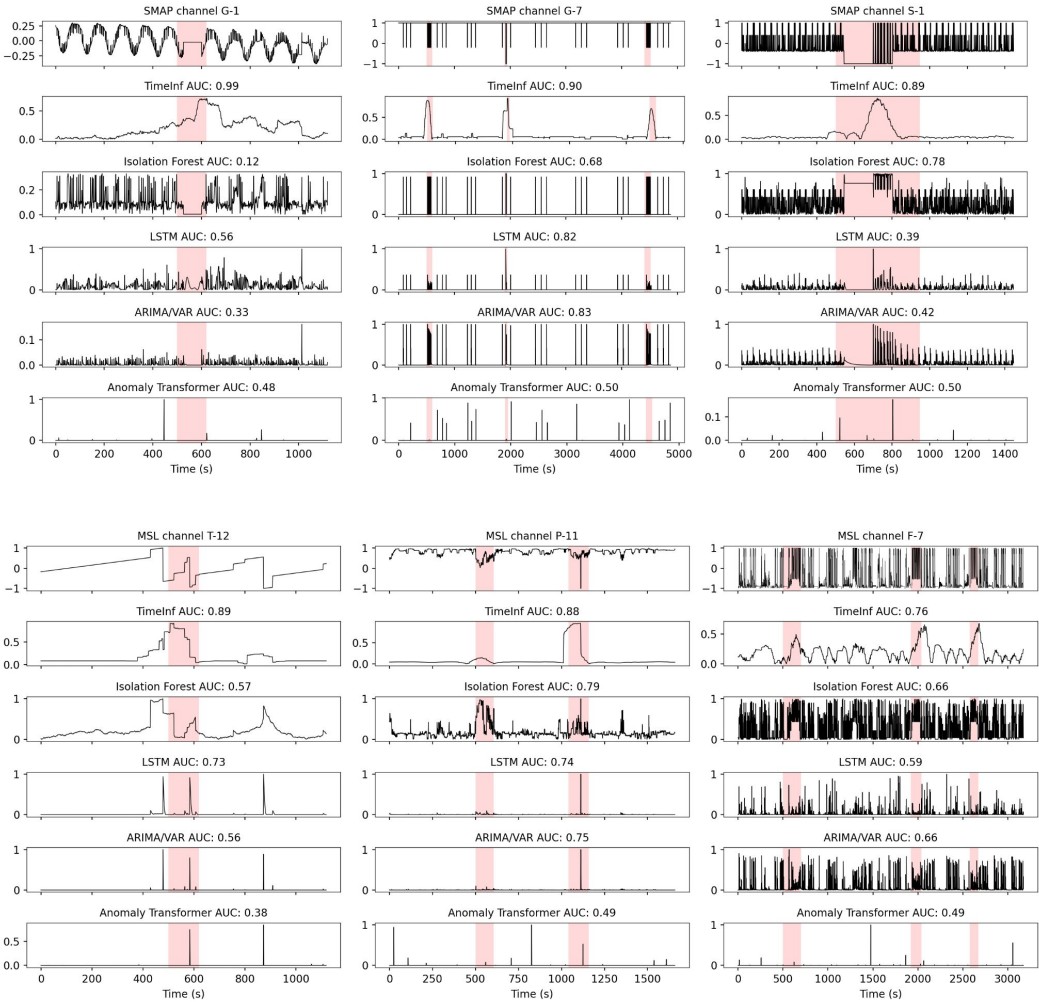

Figure 10: *Qualitative results for TimeInf and other baseline anomaly detectors on selected channels from the SMAP and MSL datasets.* The first row of each panel displays the first dimension of the time series, which captures the major anomaly pattern in the data. The remaining rows of each panel show the anomaly scores generated by each detector. For better visualizations, we normalize the anomaly scores from each method to a range between 0 and 1. The ground truth anomaly intervals are marked in red segments. TimeInf outperforms state-of-the-art time series anomaly detection methods in identifying both single and multiple anomaly patterns in the data.

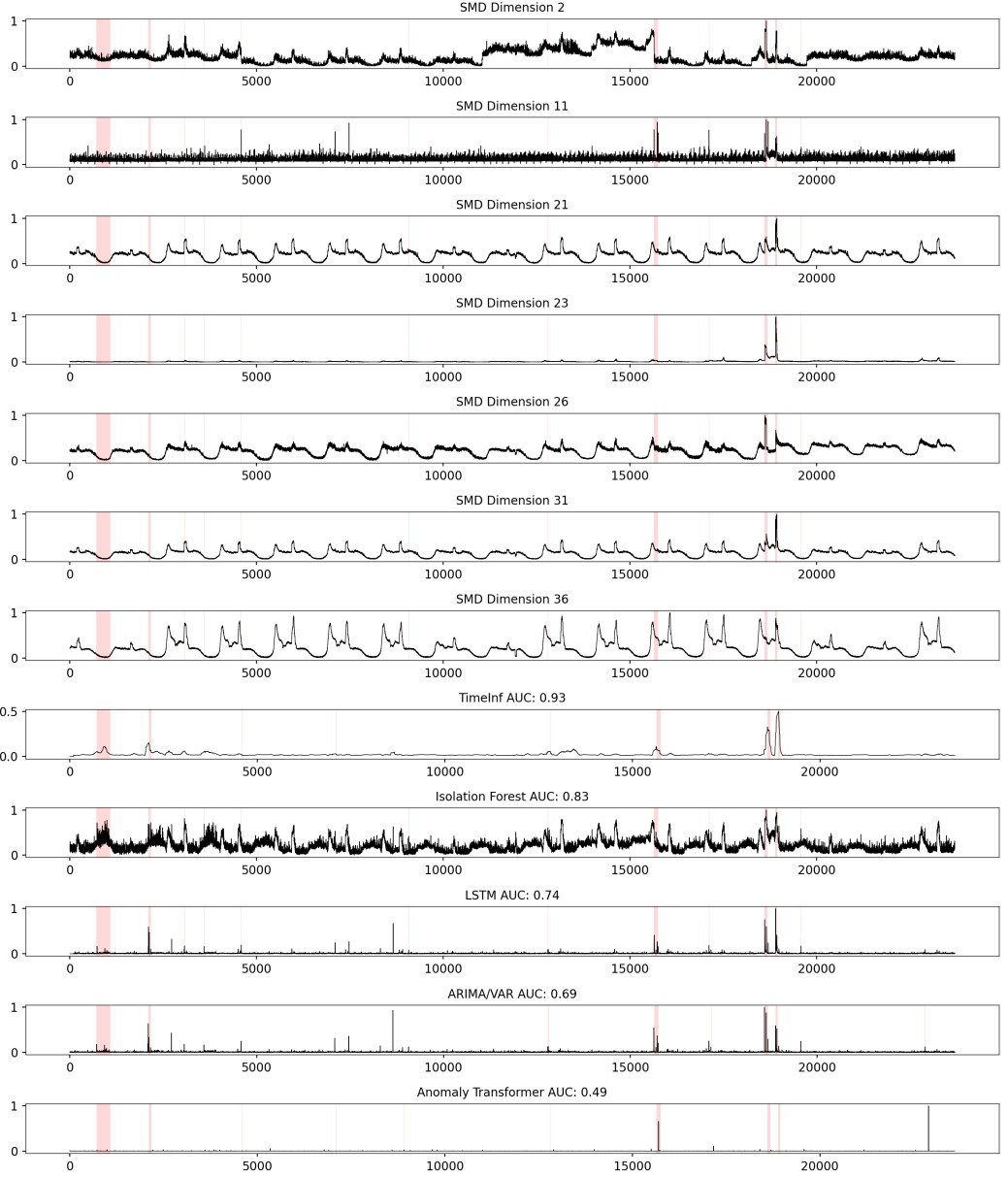

Figure 11: *Qualitative results for TimeInf and other baseline anomaly detectors on a selected channel from the multivariate SMD dataset.* The first seven rows show selected dimensions of the time series, while the remaining rows show the anomaly scores generated by each detector. For better visualizations, we normalize the anomaly scores from each method to a range between 0 and 1. The ground truth anomaly intervals are marked in red segments. TimeInf outperforms state-of-the-art time series anomaly detection methods in identifying anomaly patterns in multivariate time series data.

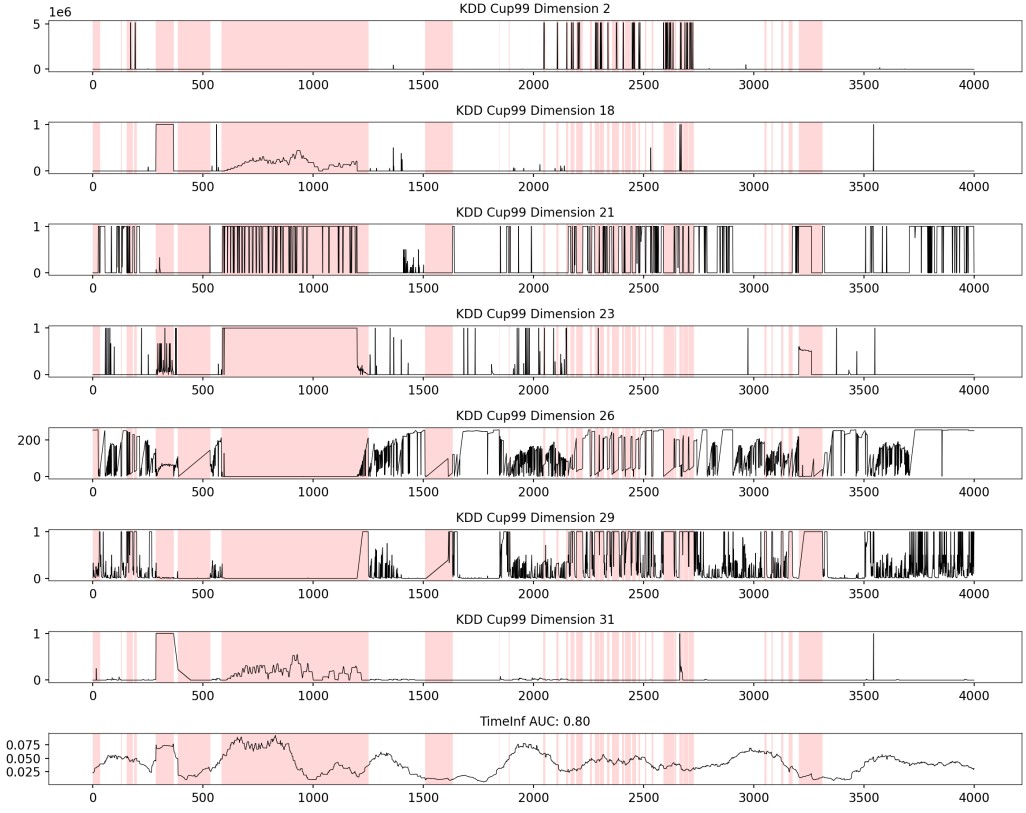

Figure 12: *A qualitative example of mislabeled anomaly points in the KDD-Cup99 dataset.* The first seven rows showcase selected dimensions of the time series, while the last row displays the anomaly scores generated by TimeInf. For better visualizations, we normalize the anomaly scores from each method to a range between 0 and 1. The ground truth anomaly intervals are marked in red segments. While not explicitly marked as anomalies, segments within the ranges 0 to 400, 1250 to 1500, and 3500 to 3600 exhibit distinct patterns that set them apart from others. TimeInf yields elevated anomaly scores for time points within these segments, raising questions about the reliability of the labels in these instances.

