# OpenReview forum: "TimeInf: Time Series Data Contribution via Influence Functions"
_ICLR.cc/2025/Conference — ICLR 2025 Poster_

### Official Review · Reviewer_Zos6 · 2024-10-21

**Soundness:** 3
**Presentation:** 2
**Contribution:** 2
**Rating:** 6
**Confidence:** 3

**Summary:**

The paper introduces TimeInf, a model-agnostic data contribution estimation method for time-series datasets. It uses influence scores to attribute predictions to individual time points, preserving temporal dependencies. TimeInf shows effectiveness in detecting anomalies, providing interpretable attributions, and identifying mislabeled anomalies in ground truth.

**Strengths:**

The paper is well-organized with clear problem formulation and concise visualization. It effectively demonstrates the utility of TimeInf across multiple tasks, providing robust results on time-series datasets.

**Weaknesses:**

1.	My main concern is that while TimeInf is presented as a general model-agnostic method, the experiments in the main text focus solely on anomaly detection. It would be better to include more different tasks to support the advantage of this general method rather than only elaborate on one task.
2.	Table 2 contains misformatted bold values that do not reflect the best performance (UCR and SMAP’s F1).
3.	The data pruning experiments use non-standard time-series forecasting metrics; it will be better if more common metrics like RMSE and MAE can be considered.

**Questions:**

1.	Can you elaborate on how TimeInf extends beyond the local historical context to capture a more global perspective as mentioned in Section 3?
2.	Following Q1, regarding Table 3, can you provide baseline comparisons to show the model’s advantages in capturing long-term dependencies?

---

> ### Author Response · Authors · 2024-11-21
> **Thank you for reviewing our paper**
>
> > 1. My main concern is that while TimeInf is presented as a general model-agnostic method, the experiments in the main text focus solely on anomaly detection. It would be better to include more different tasks to support the advantage of this general method rather than only elaborate on one task.
>
> We appreciate the reviewer’s feedback. We would like to clarify that TimeInf is a general data valuation method that can be applied to various time series problems, and thus *it is not limited to the anomaly detection task*. To be more specific, TimeInf can be used to detect abnormal data points in the anomaly detection task when it measures the impact of a time point on itself, but it can also be used to identify influential time points in the forecasting task when it measures the impact of a time point on a particular model prediction. In Appendix E "Data Pruning" of the submitted paper, we actually demonstrated how TimeInf can be used to reduce sample size while not sacrificing model performance too much in the forecasting task. We believe there would be more impactful applications, but these two tasks serve as representative applications of TimeInf. We have included a detailed description of each task in Appendix H "Using TimeInf for Various Downstream Tasks".
>
> > 2. Table 2 contains misformatted bold values that do not reflect the best performance (UCR and SMAP’s F1).
>
> We greatly thank the reviewer for bringing this to our attention. We have corrected this error in the revised paper.
>
> > 3. The data pruning experiments use non-standard time-series forecasting metrics; it will be better if more common metrics like RMSE and MAE can be considered.
>
> We appreciate the reviewer’s suggestion and have included RMSE as a metric for data pruning in Appendix Figure 7. Like R-squared, we find that removing helpful time points identified by our method leads to a steeper rise in RMSE (indicating a decline in predictive performance) than other data valuation methods. This suggests that TimeInf is more effective in identifying beneficial points. Please refer to Appendix  E “Data Pruning” for more details.
>
> > 4. Can you elaborate on how TimeInf extends beyond the local historical context to capture a more global perspective as mentioned in Section 3?
>
> We thank the reviewer for this question. TimeInf extends beyond the local historical context by *leveraging multiple relevant time windows that include the time point of interest*. TimeInf assesses the impact of individual points under various temporal contexts, and thus some time windows even include time data points observed after the time point of interest due to its design. In contrast, traditional methods (e.g., conditional influence [1]) only consider a single time window with the previous $m-1$ time points. As a result, it becomes limited to a specific historical time window and does not account for future data points. In this sense, we claim that TimeInf extends beyond the local historical context. We will include this discussion in the revision.
>
> > 5. Following Q1, regarding Table 3, can you provide baseline comparisons to show the model’s advantages in capturing long-term dependencies?
>
> We thank the reviewer for this question. In Figure 5 of the submitted paper, we compared how the performance of TimeInf and baseline methods changes across different time window sizes in the anomaly detection task. As shown in Appendix Figure 5 Panel A, the AUC of TimeInf significantly increases as the time window size increases for most datasets, while that of other methods does not show improvement. That is, TimeInf effectively captures long-term dependencies in various datasets, which leads to improvement in anomaly detention ability, but baseline methods are less effective in capturing long-term dependencies.
>
> [1] Kunsch, H. "Infinitesimal robustness for autoregressive processes."

---

> ### Author Response · Authors · 2024-11-25
> **Follow-up**
>
> Dear Reviewer Zos6,
>
> We wanted to reach out to see if we have addressed your concerns. Please let us know if you have any questions or feedback.
>
> Thank you for your time and input!

---

> > ### Comment · Reviewer_Zos6 · 2024-11-25
> >
> > Thank you for your response. All my concerns were addressed and I raised my score to 6.

---

> > > ### Author Response · Authors · 2024-11-25
> > > **Thank you**
> > >
> > > Dear Reviewer Zos6,
> > >
> > > Glad to hear we have addressed all your concerns. Thanks again for carefully reviewing and re-evaluating our paper.
> > >
> > > Best,
> > > Authors

---

### Official Review · Reviewer_kAP5 · 2024-11-01

**Soundness:** 2
**Presentation:** 2
**Contribution:** 3
**Rating:** 6
**Confidence:** 3

**Summary:**

This work presents TimeInf, a data contribution estimation model for time series. TimeInf overcomes the traditional i.i.d. setting of influence functions, making it more suitable for time series data. By considering multiple time blocks, TimeInf captures contextual information within time series data. Additionally, its model-agnostic design enhances its versatility and flexibility.

**Strengths:**

+ TimeInf addresses the gap between existing data contribution methods and the unique requirements of time series data.
+ The proposed influence function is highly versatile, with multiple variants supporting diverse anomaly detection settings. Its model-agnostic nature further enhances its applicability across various models.
+ TimeInf is computationally efficient, leading to significant resource savings.

**Weaknesses:**

+ Some key mathematical terms in the paper could benefit from clearer explanations, such as $\delta_{x}[m]$ and $z^[m]$.
+ The paper lacks a clear mathematical definition of the task, especially regarding the tasks targeted by the variants of TimeInf, which could cause confusion.
+ Although the authors describe TimeInf as a data contribution estimation method, its task definition and benchmark choices make it appear primarily as an anomaly detection model. While data pruning experiments are provided in Appendix Section E, more direct evidence would help demonstrate that TimeInf goes beyond anomaly detection.
+ The comparisons would benefit from including more advanced anomaly detection algorithms [1], [2], [3].
+ In Section 4.3, the authors attribute TimeInf's relatively poor performance in Table 4 to mislabeled ground truth annotations. However, they need to clarify why other benchmark models are unaffected by these mislabeled annotations. Demonstrating that other models do adhere to potentially mislabeled annotations would help validate this claim.

**Questions:**

+ Experimentally, TimeInf can effectively identify anomalies in time series. However, in the Data Pruning experiment, the authors claim that TimeInf can identify the most influential time patterns, which appears different from detecting anomalies (removing anomalies should not necessarily lead to a significant drop in model performance). Why is TimeInf able to identify influential time patterns at times, while it detects anomalies at other times? This seems somewhat contradictory.

[1]. Tuli S, Casale G, Jennings N R. TranAD: deep transformer networks for anomaly detection in multivariate time series data[J]. Proceedings of the VLDB Endowment, 2022, 15(6): 1201-1214.

[2]. Xiao C, Gou Z, Tai W, et al. Imputation-based time-series anomaly detection with conditional weight-incremental diffusion models[C]//Proceedings of the 29th ACM SIGKDD Conference on Knowledge Discovery and Data Mining. 2023: 2742-2751.

[3]. Xu H, Wang Y, Jian S, et al. Calibrated one-class classification for unsupervised time series anomaly detection[J]. IEEE Transactions on Knowledge and Data Engineering, 2024.

---

> ### Author Response · Authors · 2024-11-21
> **Thank you for reviewing our paper**
>
> > 1. Some key mathematical terms in the paper could benefit from clearer explanations, such as \delta_x^{[m]} and z^{[m]}.
>
> We thank the reviewer for the suggestion. The term $z^{[m]}$ is defined in Section 2 as a block of m consecutive time points, represented as an m-dimensional vector. Additionally, $\delta_x^{[m]} = P(X^{[m]} = x^{[m]}) = 1$ is defined in Section 2 as a point mass distribution that assigns all the probability mass to a single time block. This means that the contaminating distribution must take the specific form of the observed time block, thereby eliminating the possibility of the time block assuming other values. We use this notation consistently throughout the paper, but please let us know if you have any further concerns.
>
> > 2. The paper lacks a clear mathematical definition of the task, especially regarding the tasks targeted by the variants of TimeInf, which could cause confusion.
>
> We appreciate the reviewer’s suggestion. To clarify how to use TimeInf in both anomaly detection and forecasting tasks, we have added Appendix H “Using TimeInf for Various Downstream Tasks”. Please refer to that section for additional details.
>
> > 3. Although the authors describe TimeInf as a data contribution estimation method, its task definition and benchmark choices make it appear primarily as an anomaly detection model. While data pruning experiments are provided in Appendix Section E, more direct evidence would help demonstrate that TimeInf goes beyond anomaly detection.
>
> We appreciate the reviewer’s feedback. TimeInf is a general data valuation method that assesses the contribution of each time point to model predictions, and its application is not limited to anomaly data detection. While we focused on its effectiveness in identifying harmful time points through anomaly detection, TimeInf can also identify beneficial time points in forecasting, as detailed in Appendix E "Data Pruning" of the submitted paper. These tasks serve as representative examples of TimeInf's capability to identify both detrimental and beneficial time points.  If the reviewer has any specific applications in mind, we would appreciate your suggestions and would be happy to explore them.
>
> > 4. The comparisons would benefit from including more advanced anomaly detection algorithms [1], [2], [3].
>
> We thank the reviewer for suggesting additional anomaly detection methods. In response, we conducted additional experiments to compare our method against two state-of-the-art transformer-based approaches: TranAD [1] and DCdetector [2]. We have updated Table 2 in the paper to reflect these results. As shown, TimeInf outperforms TranAD by $7 \sim~ 30$% in AUC and DCdetector by $3 \sim 17$% in AUC across the UCR, SMAP, MSL, NAB-Traffic, and SMD datasets. We have updated Table 2 and included this discussion in the revision.
>
> > 5. In Section 4.3, the authors attribute TimeInf's relatively poor performance in Table 4 to mislabeled ground truth annotations. However, they need to clarify why other benchmark models are unaffected by these mislabeled annotations. Demonstrating that other models do adhere to potentially mislabeled annotations would help validate this claim.
>
> We thank the author for the suggestion. To investigate this issue, we have updated Figures 2 and 3 to include anomaly scores from Isolation Forest, which performs well on problematic datasets like PSM, SWaT, and NAB-Tweets in Appendix Table 4. However, when examining the ground truth annotations alongside the anomaly scores, some discrepancies arise. For example, Figure 2 segments 1 and 2 contain a large anomalous section that is incorrectly labeled; most of it is normal, with only a small anomalous segment. Segment 3 shows a clear anomaly between time points 5000 and 6000 but lacks an anomaly label. Isolation Forest adheres to these mislabeled annotations, and obtains a high AUC, while TimeInf accurately detects anomalies but receives a lower AUC. As a result, Isolation Forest has a higher AUC across these problematic datasets in Appendix Table 4. However, this comparison is not meaningful due to the issues with labeling in these datasets, making the evaluation unreliable. We have included this discussion in Section 4.3 of the paper.
>
> [1] Tuli, Shreshth, Giuliano Casale, and Nicholas R. Jennings. "Tranad: Deep transformer networks for anomaly detection in multivariate time series data."
>
> [2] Yang, Yiyuan, et al. "DCdetector: Dual attention contrastive representation learning for time series anomaly detection."

---

> ### Author Response · Authors · 2024-11-21
> **Thank you for reviewing our paper (continued)**
>
> > Experimentally, TimeInf can effectively identify anomalies in time series. However, in the Data Pruning experiment, the authors claim that TimeInf can identify the most influential time patterns, which appears different from detecting anomalies (removing anomalies should not necessarily lead to a significant drop in model performance). Why is TimeInf able to identify influential time patterns at times, while it detects anomalies at other times? This seems somewhat contradictory.
>
> We appreciate the reviewer’s question. We would like to clarify that TimeInf can be used to detect abnormal data points in the anomaly detection task when it measures the impact of a time point on itself, but it can also be used to identify influential time points in the forecasting task when it measures the impact of a time point on a particular model prediction. We use TimeInf to capture different quantities depending on the application. To further explain how TimeInf can be used to identify helpful points for forecasting and harmful points for anomaly detection, we have added Appendix H “Using TimeInf for Various Downstream Tasks”. Please refer to that section for more details.

---

> > ### Comment · Reviewer_kAP5 · 2024-11-24
> > **Official Comment by Reviewer kAP5**
> >
> > Thank you for your efforts in addressing most of my concerns. I have updated my score to 6.

---

> > > ### Author Response · Authors · 2024-11-24
> > > **Thank you**
> > >
> > > We are glad that we have addressed most of your concerns. Thank you for your valuable suggestions and time.

---

### Official Review · Reviewer_kJ9U · 2024-11-04

**Soundness:** 3
**Presentation:** 3
**Contribution:** 2
**Rating:** 6
**Confidence:** 4

**Summary:**

This paper introduces TimeInf, a novel method for estimating data contribution in time series datasets using influence functions. Unlike previous data contribution methods that primarily focus on i.i.d. settings, TimeInf specifically addresses the challenges of temporal dependencies in time series data. The method works by analyzing overlapping blocks of consecutive time points to preserve temporal structure, then uses influence functions to measure how individual time points impact model predictions. TimeInf's key innovation is its ability to integrate data values across consecutive time series observations while accounting for different temporal arrangements of data points. The authors demonstrate TimeInf's effectiveness through extensive experiments on real-world datasets, showing it outperforms existing methods in both identifying harmful anomalies and helpful time points for forecasting. The method is particularly noteworthy for its computational efficiency, interpretable results through visualizations, and strong performance in practical applications, while being theoretically grounded in robust statistics and autoregressive models.

**Strengths:**

The primary technical strength of TimeInf lies in its novel and theoretically sound approach to data contribution estimation in time series. It successfully extends influence functions to handle temporal dependencies, addressing a significant gap in time series analysis that previous methods overlooked. The method is built on a strong mathematical foundation combining robust statistics with innovative use of overlapping blocks to preserve temporal structure. This theoretical rigor is balanced with practical efficiency, as TimeInf proves to be computationally faster than deep learning alternatives while remaining scalable to large datasets with millions of time points. The method provides clear, quantifiable influence scores and intuitive visualizations that help users understand both anomaly patterns and forecasting behaviors. Its comprehensive evaluation across multiple real-world datasets demonstrates consistent improvement over existing approaches in both effectiveness and efficiency.

**Weaknesses:**

1. Methodological Limitations:
The paper’s core claim about their “distinctive integration” considering various temporal patterns seems to overlap significantly with existing attention mechanisms, without adequately differentiating itself. The theoretical justification for TimeInf lacks rigorous analysis of its properties, such as consistency or asymptotic behavior. Additionally, the paper doesn’t thoroughly explore the sensitivity of TimeInf to various hyperparameters, like block length or model architecture choices.
	2.	Experimental Issues:
The experimental setup, particularly for anomaly detection tasks, is not clearly described, making result reproduction challenging. The reported results differ significantly from those in original papers for baseline methods, raising questions about comparison fairness. Important baselines in time series anomaly detection such as MTAD-GAT[1], TransAD[2] are missing, limiting the ability to fully assess TimeInf’s performance relative to state-of-the-art methods.  [1] and [2] are using the unspervised learning to learn the distribution and don't use the "clean data assumption" as well. The paper also lacks a comprehensive evaluation of TimeInf’s interpretability claims compared to other approaches.
	3.	Limited Scope and Analysis:
While the paper focuses primarily on anomaly detection tasks, it’s not clear how well TimeInf generalizes to other time series tasks or domains. The computational complexity of TimeInf compared to existing approaches is not thoroughly analyzed, despite mentioning efficiency improvements through conjugate gradient and Hessian-free approaches. The paper lacks a thorough discussion of TimeInf’s limitations or potential failure cases, which is crucial for understanding when and where the method is most applicable. Lastly, the evaluation metrics used are standard (AUC and F1 score), but don’t consider other potentially relevant metrics for time series anomaly detection, such as detection delay or false positive rate at a fixed detection threshold.


[1] Zhao, Hang, et al. "Multivariate time-series anomaly detection via graph attention network." 2020 IEEE international conference on data mining (ICDM). IEEE, 2020.
[2] Tuli, Shreshth, Giuliano Casale, and Nicholas R. Jennings. "TranAD: deep transformer networks for anomaly detection in multivariate time series data." Proceedings of the VLDB Endowment 15.6 (2022): 1201-1214.

**Questions:**

See weakness.

---

> ### Author Response · Authors · 2024-11-21
> **Thank you for reviewing our paper**
>
> > 1. Methodological Limitations: The paper’s core claim about their “distinctive integration” considering various temporal patterns seems to overlap significantly with existing attention mechanisms, without adequately differentiating itself.
>
> We would like to thank the reviewer for the feedback. Regarding the comment that our method “overlaps significantly with existing attention mechanisms,” we believe our approach operates differently from traditional attention mechanisms. Attention scores assess the contribution of each timestep (or token) **within a specific time window** to a model's prediction, which is somewhat similar to the conditional influence method [1] introduced in previous work. However, our proposed method also considers the various potential contexts of a time point when evaluating its impact on model predictions.
>
> > 2. The theoretical justification for TimeInf lacks rigorous analysis of its properties, such as consistency or asymptotic behavior.
>
> We thank the reviewer for bringing this to our attention. We agree that analyzing the theoretical properties of TimeInf is an important research topic. However, the consistency and asymptotic properties of statistics are often studied in specific contexts (e.g., linear models and i.i.d. data) while the proposed method is model-agnostic and applies to various time series problems. In addition, these theoretical properties often require in-depth mathematical analysis, which we believe is crucial in the literature but is beyond the scope of this paper.
>
> > 3. Additionally, the paper doesn’t thoroughly explore the sensitivity of TimeInf to various hyperparameters, like block length or model architecture choices.
>
> We would like to clarify that we have explored the sensitivity of TimeInf to hyperparameters, such as block length and model architecture choices, in Appendix D “Ablation Study” of the submitted paper. Our analysis shows that while the performance of TimeInf improves with increasing block length, it becomes less sensitive once the block length exceeds 100. Additionally, we found that model choices do impact performance, with both linear AR models and transformer-based models producing highly accurate influence scores.
>
> > 4. Experimental Issues: The experimental setup, particularly for anomaly detection tasks, is not clearly described, making result reproduction challenging.
>
> We appreciate the reviewer’s feedback. Our supplementary materials include the code and a README.md file, ensuring that the results are reproducible. Furthermore, we have revised the experimental setup in the paper to improve reproducibility.
>
> > 5. The reported results differ significantly from those in original papers for baseline methods, raising questions about comparison fairness.
>
> We want to clarify that the reported results in Table 2 differ from those in original papers because a different experimental setting is used, and we have explicitly explained this difference in Section 4.1 of the submitted paper. To summarize the main difference, our experimental setting considers realistic scenarios where the training dataset is NOT anomaly-free while conventional time series anomaly detection settings typically consider well-curated, clean training datasets. Due to this difference, our result differs from the original work. Please see Section 4.1 of the submitted paper for detailed comparisons.
>
> > 6. Important baselines in time series anomaly detection such as MTAD-GAT[1], TransAD[2] are missing, limiting the ability to fully assess TimeInf’s performance relative to state-of-the-art methods. [1] and [2] are using unsupervised learning to learn the distribution and don't use the "clean data assumption" as well.
>
> We greatly thank the reviewer for suggesting state-of-the-art anomaly detection methods. In response, we conducted additional experiments to compare our method against two transformer-based approaches: TranAD [2] and DCdetector [3]. We have updated Table 2 in the paper to reflect these results. As shown, TimeInf outperforms TranAD by $7\sim30$% in AUC and DCdetector by $3\sim17$% in AUC across the UCR, SMAP, MSL, NAB-Traffic, and SMD datasets.
>
> > 7. The paper also lacks a comprehensive evaluation of TimeInf’s interpretability claims compared to other approaches.
>
> Regarding the evaluation of TimeInf’s interpretability, we provide a visual comparison of the anomaly scores from TimeInf versus other baselines in Figure 1 of the paper, showing that TimeInf assigns interpretable scores to time points across different contexts. If this does not fully address your question, please let us know.
>
> [1] Kunsch, H. "Infinitesimal robustness for autoregressive processes."
>
> [2] Tuli, Shreshth, Giuliano Casale, and Nicholas R. Jennings. "Tranad: Deep transformer networks for anomaly detection in multivariate time series data."
>
> [3] Yang, Yiyuan, et al. "DCdetector: Dual attention contrastive representation learning for time series anomaly detection."

---

> ### Author Response · Authors · 2024-11-21
> **Thank you for reviewing our paper (continued)**
>
> > 8. Limited Scope and Analysis: While the paper focuses primarily on anomaly detection tasks, it’s not clear how well TimeInf generalizes to other time series tasks or domains.
>
> We appreciate the reviewer’s feedback. TimeInf is a general data valuation method that assesses the contribution of each time point to model predictions, not limited to anomaly detection. Although we focus on its effectiveness in identifying harmful time points through anomaly detection in the main paper, TimeInf can also identify beneficial time points in forecasting. We detail this application in Appendix E “Data Pruning” due to page limitations. To further clarify how to use TimeInf in both anomaly detection and forecasting tasks, we have added Appendix H “Using TimeInf for Various Downstream Tasks”. If the reviewer has any specific applications in mind, we would be happy to explore how TimeInf could be applied to those scenarios.
>
> > 9. The computational complexity of TimeInf compared to existing approaches is not thoroughly analyzed, despite mentioning efficiency improvements through conjugate gradient and Hessian-free approaches.
>
> We thank the reviewer for pointing out this issue. It is quite challenging to fairly compare the computational complexity between different methods because they depend on specific models or training algorithms. Because of this reason, instead of a theoretical comparison, we conducted an empirical comparison in Table 2 of the submitted paper by measuring the total runtime used in the anomaly detection task. Based on the results, TimeInf is significantly faster than most methods, with the exceptions of Isolation Forest, Conditional Influence, and TranAD. However, TimeInf outperforms these methods in anomaly detection while maintaining a short run-time.
>
> > 10. The paper lacks a thorough discussion of TimeInf’s limitations or potential failure cases, which is crucial for understanding when and where the method is most applicable.
>
> In Section 5 “Concluding Remarks,” we explicitly discussed the limitations of TimeInf. We found that the performance improvements of TimeInf over other data contribution methods are marginal in high-dimensional data. Additionally, the time complexity of TimeInf is quadratic with respect to the dimensionality of the time series, resulting in higher computational costs in high-dimensional settings.
>
> > 11. Lastly, the evaluation metrics used are standard (AUC and F1 score), but don’t consider other potentially relevant metrics for time series anomaly detection, such as detection delay or false positive rate at a fixed detection threshold.
>
> We appreciate the reviewer’s suggestion for additional metrics in anomaly detection evaluation. While we did not consider detection delay, we do account for the false positive rate (FP), as it is inherently included in the computation of the AUC.

---

> ### Author Response · Authors · 2024-11-25
> **Follow-up**
>
> Dear Reviewer kJ9U,
>
> Thank you once again for your time and constructive reviews. We wanted to reach out to see if you have any remaining concerns or comments. Please let us know if you have any questions or feedback!
>
> Best,
> Authors

---

> > ### Comment · Reviewer_kJ9U · 2024-11-25
> > **Thanks for the response**
> >
> > Dear authors,
> >
> > Thanks for your response. I think most of my concerns have been addressed. I will maintain my original scores.
> >
> > Best

---

> > > ### Author Response · Authors · 2024-11-26
> > > **Thank you!**
> > >
> > > Dear Reviewer kJ9U,
> > >
> > > Glad to know we successfully addressed all your concerns. Thank you once again for many helpful comments.
> > >
> > > Best,
> > > Authors

---

### Official Review · Reviewer_xCEM · 2024-11-05

**Soundness:** 3
**Presentation:** 3
**Contribution:** 3
**Rating:** 6
**Confidence:** 2

**Summary:**

This paper presents a new method for measuring the contribution of different timestamp to the final predictions. The major idea is to leverage influence scores to assign model predictions to individual time points while preserving temporal structures between the time points. Experiments in anomaly detection can verify the effectiveness of the proposed method.

**Strengths:**

1. The paper is well-written and easy to follow.
2. The presented solution is technically solid.
3. The experiments on anomaly detection can demonstrate the efficacy of the proposed method.

**Weaknesses:**

1. The relationship between the proposed model and unsupervised anomaly detection is unclear and should be further explained.

2. The model's generalization ability to other time series tasks (e.g., forecasting, imputation) is uncertain. Specifically, how would the contribution score of historical timestamps be measured for future time points?

3. There is no discussion on scalability. How does the method perform as the number of historical time points increases?

**Questions:**

Please reply to the above weaknesses if I have any misunderstandings. Thank you.

Another question: Can the proposed model process multivariate time series?

---

> ### Author Response · Authors · 2024-11-21
> **Thank you for reviewing our paper**
>
> > 1. The relationship between the proposed model and unsupervised anomaly detection is unclear and should be further explained.
>
> We appreciate the reviewer’s feedback and would like to clarify that our work introduces a general data valuation method that assigns a score to each data point, evaluating its impact on the model’s reconstruction quality. Our approach can identify both harmful points, which are relevant for anomaly detection, and beneficial time points that contribute to prediction quality. Our method has a broader scope; while we show its application in anomaly detection, it can be used for other tasks, such as identifying helpful data points for forecasting. In comparison, unsupervised anomaly detection methods are more specialized and focused only on anomaly detection. To clarify how TimeInf can be used for both anomaly detection and forecasting, we have added Appendix H "Using TimeInf for Various Downstream Tasks". Please refer to that section for more details.
>
> > 2. The model's generalization ability to other time series tasks (e.g., forecasting, imputation) is uncertain. Specifically, how would the contribution score of historical timestamps be measured for future time points?
>
> We thank the reviewer for this question. The proposed TimeInf can be used to assess a historical time point’s contribution to a time series prediction in forecasting problems. We have explicitly stated this in Appendix E "Data Pruning" of the submitted paper. To further clarify how to use TimeInf in a forecasting task, we have added Appendix H "Using TimeInf for Various Downstream Tasks". Please refer to that section for more details.
>
> > 3. There is no discussion on scalability. How does the method perform as the number of historical time points increases?
>
> We provide an analysis of computational complexity in Section 3.2 of the submitted paper. The complexity of TimeInf computation is $O(nq^2 + q^3)$, where $n$ is the number of data points and $q$ denotes the dimensionality of the data, particularly when using the conjugate gradient (CG) for Hessian approximation. In other words, it increases linearly with the number of historical time points.
>
> > Another question: Can the proposed model process multivariate time series?
>
> Yes, the proposed method is applicable to multivariate time series, and in fact, the SMD anomaly detection dataset we presented in Section 4.1 of the submitted paper is a multivariate dataset. In Section 4.1 “TimeInf Usage,” we have explicitly explained the details of its application to multivariate datasets.

---

> > ### Comment · Reviewer_xCEM · 2024-11-22
> > **Feedback**
> >
> > Dear authors,
> >
> > Thank you for addressing my questions. I believe it would be beneficial for the paper to explore more general tasks in time series analysis. While the paper received some negative feedback from other reviewers, I would like to maintain my recommendation score due to its technical robustness and effectiveness.
> >
> > Reviewer

---

> > > ### Author Response · Authors · 2024-11-22
> > > **Thank you**
> > >
> > > We are glad that we have addressed your questions. Thank you for your valuable suggestions and time.

---

### Official Review · Reviewer_rAep · 2024-11-08

**Soundness:** 2
**Presentation:** 3
**Contribution:** 2
**Rating:** 5
**Confidence:** 3

**Summary:**

This paper presents an interesting approach to time series interpretability research—by using noise perturbations to measure the contribution of data to the model. However, the method designed by the authors is a theoretical extension of existing approaches and lacks more extensive experimental evidence to demonstrate its effectiveness.

**Strengths:**

The idea of inferring interpretability and anomaly analysis by measuring the contribution of time series data to the model's output is quite novel. The mathematical derivation is well-developed, and the details are thorough.

**Weaknesses:**

1. Compared to existing methods, the advantages of this approach are not fully emphasized, and the innovative aspects are not clearly defined. The main contribution of the paper can be seen as an extension of existing theory, but the method primarily involves deriving formulas based on Equation (3) and considering the Mixture of \( \delta z^{[m]} \). This operation has limited significance for theoretical extension and does not effectively demonstrate the contribution claimed. I do not believe it addresses the shortcomings of existing methods.
2. The paper conducts experiments on anomaly detection, interpretability, and mislabeled error in datasets. However, the paper does not discuss current state-of-the-art baselines in anomaly detection, such as methods based on LLMs or DCdetector. Moreover, the AUC and F1 results differ significantly from the original work. While the authors' skepticism about dataset quality is commendable, more evidence is needed to support this claim, as only partial visualizations of the data are presented. Additionally, the ablation experiments in the appendix are limited to parameter experiments. Based on my experience, complex theoretical methods often perform unsatisfactorily in neural network training, and more evidence is required to validate the effectiveness of the theory.
3. Compared to traditional machine learning papers, there is no clear and complete process to present the designed algorithm, such as the input, output, and objective function. The majority of the paper is focused on theoretical formulas and related comparisons, making it difficult to understand

**Questions:**

See above.

---

> ### Author Response · Authors · 2024-11-21
> **Thank you for reviewing our paper**
>
> > 1. Compared to existing methods, the advantages of this approach are not fully emphasized, and the innovative aspects are not clearly defined.  The main contribution of the paper can be seen as an extension of existing theory, but the method primarily involves deriving formulas based on Equation (3) and considering the Mixture of ( \delta z^{[m]} ). This operation has limited significance for theoretical extension and does not effectively demonstrate the contribution claimed. I do not believe it addresses the shortcomings of existing methods.
>
> We thank the reviewer for their feedback and acknowledge their point that our method involves deriving new formulas based on Equation (3) and using a mixture as a contaminating distribution. However, we would like to emphasize that our work is not a simple extension of existing theory. Equation (3) resembles the regular influence function [1], but it introduces important modifications that address the limitations of the regular influence function. The regular influence function is defined for m = 1 (a time window of length 1) within the general influence function framework presented in Equation (1). This limitation restricts its use in time series analysis, where, for instance, an AR(1) model requires two consecutive time points, making the m = 1 condition impractical (See Example 2 of the submitted paper). Since Equation (3) captures different quantities compared to the regular influence, we have provided a detailed derivation in Appendix A “Deriving Influence Functions for M-Estimators.”
>
> Additionally, the new formula we derived is not just a plug-in method; it has a theoretical foundation. In Equation (7), we show that TimeInf can be interpreted as the limit of the expected infinitesimal change resulting from a data perturbation sampled from the mixture distribution. Unlike existing influence function variations, the expectation in Equation (7) captures the real impact of a single time point by accounting for the various contexts in which it can occur, thus avoiding the randomness associated with a specific time window context.
>
> > 2. The paper conducts experiments on anomaly detection, interpretability, and mislabeled error in datasets. However, the paper does not discuss current state-of-the-art baselines in anomaly detection, such as methods based on LLMs or DCdetector.
>
> We thank the reviewer for suggesting additional anomaly detection methods. In response, we have incorporated TranAD [2] and DCdetector [3] into our anomaly detection experiments.  TimeInf outperforms TranAD by $7 \sim 30$\% in AUC and DCdetector by $3 \sim17$\% in AUC across the UCR, SMAP, MSL, NAB-Traffic, and SMD datasets. These results are now included in Table 2 of the paper.
>
> > 3. Moreover, the AUC and F1 results differ significantly from the original work.
>
> We want to clarify that the AUC and F1 scores in Table 2 differ from the original work because a different experimental setting is used, and we have explicitly explained this difference in Section 4.1 of the submitted paper. To summarize the main difference, our experimental setting considers realistic scenarios where the training dataset is NOT anomaly-free while conventional time series anomaly detection settings typically consider well-curated, clean training datasets. Due to this difference, our result differs from the original work. Please see Section 4.1 of the submitted paper for detailed comparisons.
>
> [1] Koh, Pang Wei, and Percy Liang. "Understanding black-box predictions via influence functions."
>
> [2] Tuli, Shreshth, Giuliano Casale, and Nicholas R. Jennings. "Tranad: Deep transformer networks for anomaly detection in multivariate time series data."
>
> [3] Yang, Yiyuan, et al. "DCdetector: Dual attention contrastive representation learning for time series anomaly detection."

---

> ### Author Response · Authors · 2024-11-21
> **Thank you for reviewing our paper (continued)**
>
> > 4. While the authors' skepticism about dataset quality is commendable, more evidence is needed to support this claim, as only partial visualizations of the data are presented.
>
> Thank you for the suggestion. We have found critical evidence of problematic annotations through basic statistics. Our exploratory analysis shows that the anomaly labels in the PSM, NAB, and SWaT datasets are unrealistic for real-world anomaly detection. In the PSM dataset, 27.76% of the data is labeled as anomalous, raising concerns since anomalies are typically defined as “deviations from the normal or usual types”; if anomalies comprise nearly one-third of the data, their labeling is questionable. The NAB-Taxi dataset contains over 50% false negatives, where actual anomalies are incorrectly labeled as normal points in the ground truth annotations. This can distort evaluation metrics like precision and F1 score, giving a misleading picture of model performance. The SWaT dataset contains a single anomaly event lasting 35,800 time steps, which is significantly longer than other anomaly intervals, disproportionately affecting the F1 score and contributing to the high performance of various algorithms.
>
> In addition, we are not the first to highlight issues with mislabeled annotations in anomaly detection datasets. In fact, multiple studies have pointed out these problems in the SWaT, PSM, and NAB-Taxi datasets. For example, Sehili and Zhang [4], and Wagner et al. [5] both noted that “evaluations on SWaT are highly unreliable and these datasets are not suitable for time series anomaly detection assessments.” Additionally, Wu and Keogh [6] identified that “the majority of the individual examples in these datasets suffer from one or more of four flaws.” Therefore, further research is needed to thoroughly address the major issues with these problematic datasets. We plan to include this discussion in the paper in future revisions.
>
> > 5. Additionally, the ablation experiments in the appendix are limited to parameter experiments. Based on my experience, complex theoretical methods often perform unsatisfactorily in neural network training, and more evidence is required to validate the effectiveness of the theory.
>
> Thank you for your question. We would like to clarify that we did an ablation experiment in Appendix C "Model Agnosticity" to assess the impact of different model architectures on the performance of TimeInf. We evaluated TimeInf using linear, RNN, LSTM, and transformer-based PatchTST models. The results show that PatchTST performs best on the MSL dataset, while its performance is comparable to the linear AR model on the SMAP dataset. This suggests two important conclusions: first, TimeInf can achieve better performance than the one we presented in the manuscript when model architecture is carefully chosen, and second, deep neural network models are applicable to our framework and demonstrate competitive performance.
>
> > 6. Compared to traditional machine learning papers, there is no clear and complete process to present the designed algorithm, such as the input, output, and objective function. The majority of the paper is focused on theoretical formulas and related comparisons, making it difficult to understand.
>
> We thank the reviewer for the constructive suggestion. We have updated the paper to include an algorithm for TimeInf computation. Please refer to Algorithm 1 in Appendix A “Deriving Influence Functions for M-Estimators” for details.
>
> [4] Sehili, Mohamed El Amine, and Zonghua Zhang. "Multivariate time series anomaly detection: Fancy algorithms and flawed evaluation methodology."
>
> [5] Wagner, Dennis, et al. "Timesead: Benchmarking deep multivariate time-series anomaly detection."
>
> [6] Renjie Wu and Eamonn Keogh. "Current time series anomaly detection benchmarks are flawed and
> are creating the illusion of progress."

---

> ### Author Response · Authors · 2024-11-25
> **Follow-up**
>
> Dear Reviewer rAep,
>
> We wanted to reach out to see if we have addressed your concerns. Please let us know if you have any questions or feedback.
>
> Thank you for your time and input!

---

### Meta-Review · Area_Chair_3fbH · 2024-12-17

**Metareview:**

This paper proposes TimeInf, a data valuation method adapted for time-series that uses an influence-based scoring to identify the contribution of each time point to model predictions. Reviewers generally found the idea promising and the experiments convincing after revisions. Thanks for your responses!

The authors were quite active and responsive during the discussion, adding comparisons to advanced baselines like TranAD and DCdetector, clarifying differences in experimental setups, and addressing labeling issues in certain datasets. The reviewers who initially had doubts ended up mostly satisfied, with two explicitly raising their scores.

In the end, four out of five reviewers moved toward a positive stance, each scoring around marginal acceptance or better, while one remained at a weaker position due to some misunderstandings regarding the experimental setting.

Still, the authors carefully addressed each concern, showing new experiments and highlighting that differences in AUC/F1 from original references are caused by more realistic (non-clean) training scenarios. Two recurring points were that TimeInf should demonstrate broader applicability beyond anomaly detection, and clarify theoretical and setup details. The authors responded by showing how TimeInf can be used in forecasting tasks and provided more thorough definitions and derivations, which improved the sentiment of most reviewers.

With my personal view, this paper should be accepted.

**Additional Comments On Reviewer Discussion:**

see above

---

### Decision · Program_Chairs · 2025-01-22

Accept (Poster)